# Human attention during goal-directed reading comprehension relies on task optimization

Jiajie Zou[1,2], Yuran Zhang[1], Jialu Li[3], Xing Tian[3], Nai Ding[1,2]*

[1]Key Laboratory for Biomedical Engineering of Ministry of Education, College of Biomedical Engineering and Instrument Sciences, Zhejiang University, Hangzhou, China; [2]Nanhu Brain-computer Interface Institute, Hangzhou, China; [3]Division of Arts and Sciences, New York University Shanghai, Shanghai, China

**Abstract** The computational principles underlying attention allocation in complex goal-directed tasks remain elusive. Goal-directed reading, that is, reading a passage to answer a question in mind, is a common real-world task that strongly engages attention. Here, we investigate what computational models can explain attention distribution in this complex task. We show that the reading time on each word is predicted by the attention weights in transformer-based deep neural networks (DNNs) optimized to perform the same reading task. Eye tracking further reveals that readers separately attend to basic text features and question-relevant information during first-pass reading and rereading, respectively. Similarly, text features and question relevance separately modulate attention weights in shallow and deep DNN layers. Furthermore, when readers scan a passage without a question in mind, their reading time is predicted by DNNs optimized for a word prediction task. Therefore, we offer a computational account of how task optimization modulates attention distribution during real-world reading.

*For correspondence:
ding_nai@zju.edu.cn

## eLife assessment

This study provides a **valuable** contribution to the study of eye-movements in reading, revealing that attention-weights from a deep neural network show a statistically reliable fit to the word-level reading patterns of humans. Its evidence is **convincing** and strengthens a line of research arguing that attention in reading reflects task optimization. The work would be of interest to psychologists, neuroscientists, and machine learning researchers.

## Introduction

Attention profoundly influences information processing in the brain (*Posner and Petersen, 1990*; *Treisman and Gelade, 1980*; *Rayner, 1998*), and a large number of studies have been devoted to studying the neural mechanisms of attention. From the perspective of David Marr, the attention mechanism can be studied from three levels, that is, the computational, algorithmic, and implementational levels (*Marr, 1982*). At the computational level, attention is traditionally viewed as a mechanism to allocate limited central processing resources (*Kahneman, 1973*; *Franconeri et al., 2013*; *Lennie, 2003*; *Carrasco, 2011*; *Borji and Itti, 2012*). More recent studies, however, propose that attention is a mechanism to optimize task performance, even in conditions where the processing resource is not clearly constrained (*Dayan et al., 2000*; *Gottlieb et al., 2014*; *Legge et al., 2002*; *Liu and Reichle, 2010*; *Najemnik and Geisler, 2005*). The optimization hypothesis can explain the attention distribution in a range of well-controlled learning and decision-making tasks (*Najemnik and Geisler, 2005*;

*Navalpakkam et al., 2010*), but is rarely tested in complex processing tasks for which the optimal strategy is not obvious. Therefore, the computational principles that underlie the allocation of human attention during complex tasks remain elusive. Nevertheless, complex tasks are critical conditions to test whether the attention mechanisms abstracted from simpler tasks can truly explain real-world attention behaviors.

Reading is one of the most common and most sophisticated human behaviors (*Li et al., 2022*; *Gagl et al., 2022*), and it is strongly regulated by attention: Since readers can only recognize a couple of words within one fixation, they have to overtly shift their fixation to read a line of text (*Rayner, 1998*). Thus, eye movements serve as an overt expression of attention allocation during reading (*Rayner, 1998*; *Clifton et al., 2016*). Computational modeling of the eye movements has mostly focused on normal reading of single sentences. At the computational level, it has been proposed that the eye movements are programmed to, for example, minimize the number of eye movements (*Legge et al., 2002*). At the algorithmic and implementational level, models such as the E-Z reader (*Reichle et al., 2003*) can accurately predict the eye movement trajectory with high temporal and spatial resolution. Everyday reading behavior, however, often engages reading of a multiline passage and generally has a clear goal, for example, information retrieval or inference generation (*White et al., 2010*). Few models, however, have considered how the reading goal modulates reading behaviors. Here, we address this question by analyzing how readers allocate attention when reading a passage to answer a specific question in mind. The question may require, for example, information retrieval, inference generation, or text summarization (*Figure 1*). We investigate whether the task optimization hypothesis can explain the attention distribution in such goal-directed reading tasks.

Finding an optimal solution for the goal-directed reading task, however, is computationally challenging since the information related to question answering is sparsely located in a passage and their orthographic forms may not be predictable. Recent advances in DNN models, however, provide a potential tool to solve this computational problem since DNN models equipped with attention mechanisms have approached and even surpassed mean human performance on goal-directed reading tasks (*Lan et al., 2020*; *Liu et al., 2019*). Attention in DNN also functions as a mechanism to selectively extract useful information, and therefore, attention may potentially serve a conceptually similar role in DNN. Furthermore, recent studies have provided strong evidence that task-optimized DNN can indeed explain the neural response properties in a range of visual and language processing tasks (*Yamins et al., 2014*; *Kell et al., 2018*; *Goldstein et al., 2022*; *Schrimpf et al., 2021*; *Hasson et al., 2020*; *Donhauser and Baillet, 2020*; *Rabovsky et al., 2018*; *Heilbron et al., 2022*). Therefore, although the DNN attention mechanism certainly deviates from the human attention mechanism in terms of its algorithms and implementation, we employ it to probe the computational-level principle underlying human attention distribution during real-world goal-directed reading.

Here, we investigated what computational principles could generate human-like attention distribution during a goal-directed reading task. We employed DNNs to derive a set of attention weights that are optimized for the goal-directed reading task and tested whether such optimal weights could explain human attention measured by eye tracking. Furthermore, since both human and DNN processing is hierarchical, we also investigated whether the human attention distribution during different processing stages, which are characterized through different eye-tracking measures, and the DNN attention weights in different layers may be differentially influenced by visual features, text properties, and the top-down task. Additionally, we recruited both native and non-native readers to probe how language proficiency contributed to the computational optimality of attention distribution.

## Results

### Experiment 1: Task and performance

In Experiment 1, the participants (N = 25 for each question) first read a question and then read a passage based on which the question should be answered (*Figure 1A*). After reading the passage, the participants chose from four options which option was the most suitable answer to the question. In total, 800 question/passage pairs were adapted from the RACE dataset (*Lai et al., 2017*), a collection of English reading comprehension questions designed for Chinese high school students who learn English as a second language. The questions fell into six types (*Figure 1B and C*): three types of questions required attention to details, for example, retrieving a fact or generate inference based on a fact,

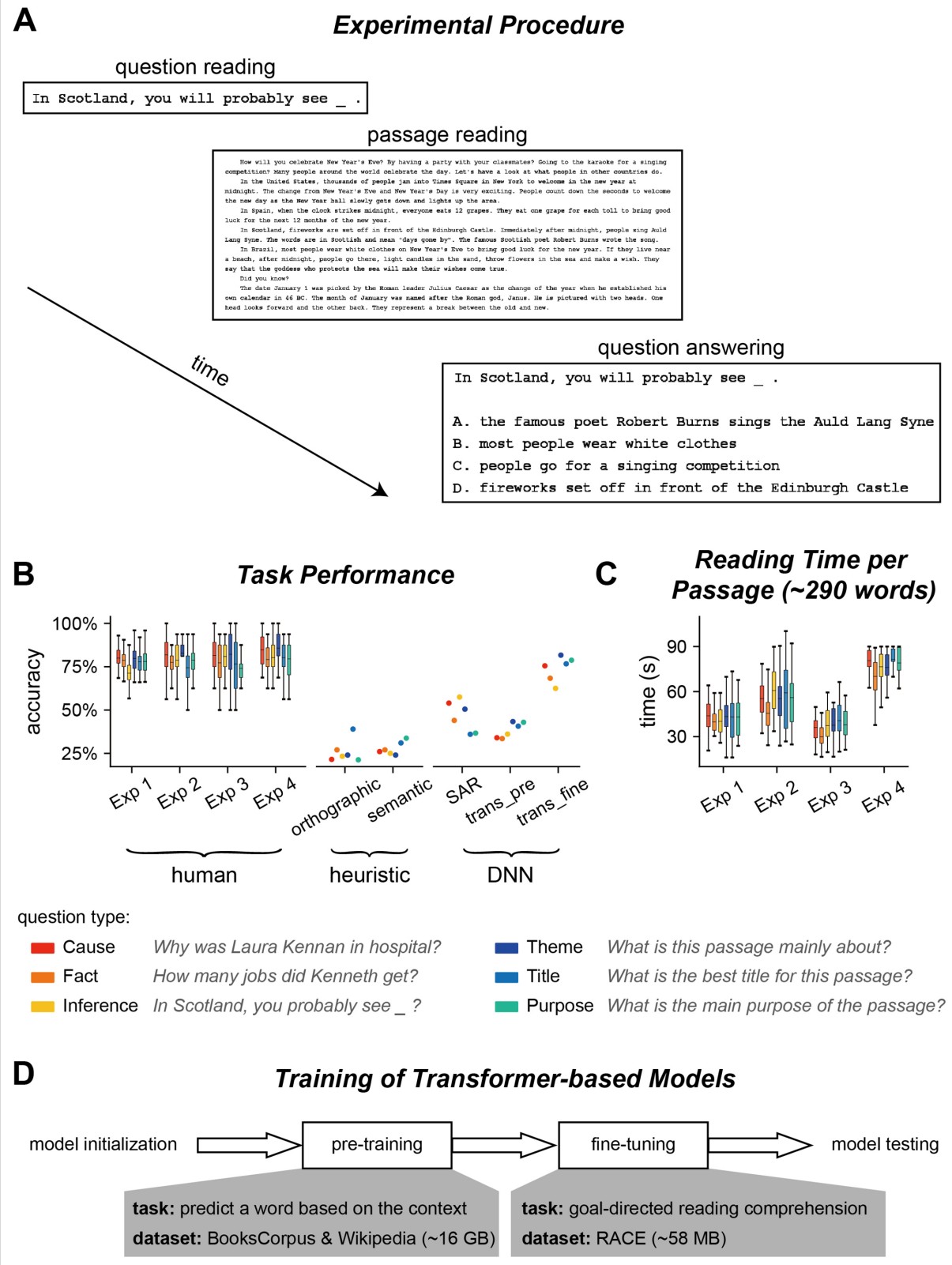

**Figure 1.** Experiment and performance. (**A**) Experimental procedure for Experiments 1–3. In each trial, participants saw a question before reading a passage. After reading the passage, they chose the answer to the question from four options. (**B**) Accuracy of question answering for humans and computational models. The question type is color coded and an example question is shown for each type. trans_pre: pre-trained transformer-based models; trans_fine: transformer-based models fine-tuned on the goal-directed reading task. (**C**) Time spent on reading each passage. The box plot

*Figure 1 continued on next page*

*Figure 1 continued*

shows the mean (horizontal lines inside the box), 25th and 75th percentiles (box boundaries), and 25th/75th percentiles ±1.5× interquartile range (whiskers) across participants (N = 25). (**D**) Illustration of the training process for transformer-based models. The pre-training process aims to learn general statistical regularities in a language based on large corpora, while the fine-tuning process trains models to perform the reading comprehension task.

The online version of this article includes the following figure supplement(s) for figure 1:

**Figure supplement 1.** Illustration of the word-level heuristic models and the recurrent neural network (RNN)-based Stanford Attentive Reader (SAR) model.

**Figure supplement 2.** Question answering accuracy for individual transformer-based models.

which were referred to as local questions. The other three types of questions concerned the general understanding of a passage, for example, summarizing the main idea or identifying the purpose of writing, which were referred to as global questions. None of the question directly appeared in the passage, and the longest string that overlapped in the passage and question was 1.8 ± 1.5 words on average.

Participants in Experiment 1 were Chinese college or graduate students who had relatively high English proficiency. The participants correctly answered 77.94% questions on average and the accuracy was comparable across the six types of questions (*Figure 1B*). We employed computational models to analyze what kinds of computations were required to answer the questions. The simplest heuristic model chose the option that best matched the passage orthographically (*Figure 1—figure supplement 1A*). This orthographic model achieved 25.6% accuracy (*Figure 1B*). Another simple heuristic model only considered word-level semantic matching between the passage and option, and achieved 27.3% accuracy (*Figure 1B*). The low accuracy of the two models indicated that the reading comprehension questions could not be answered by word-level orthographic or semantic matching.

Next, we evaluated the performance of four context-dependent DNN models, that is, Stanford Attentive Reader (SAR) (*Chen et al., 2016*), BERT (*Devlin et al., 2019*), ALBERT (*Lan et al., 2020*), and RoBERTa (*Liu et al., 2019*), which could integrate information across words to build passage-level semantic representations. The SAR used the bidirectional recurrent neural network (RNN) to integrate contextual information (*Figure 1—figure supplement 1B*) and achieved 47.6% accuracy. The other three models, that is, BERT, ALBERT, and RoBERTa, were transformer-based models that were trained in two steps, that is, pre-training and fine-tuning (*Figure 1D*). Since the three models had similar structures, we averaged the performance over the three models (see *Figure 1—figure supplement 2* for the results of individual models). The model performance on the reading task was 37.08 and 73%, respectively, after pre-training and fine-tuning (*Figure 1B*).

## Computational models of human attention distribution

In Experiment 1, participants were allowed to read each passage for 2 min. Nevertheless, to encourage the participants to develop an effective reading strategy, the monetary reward the participant received decreased as they spent more time reading the passage (see 'Materials and methods' for details). The results showed that the participants spent, on average, 0.7 ± 0.2 min reading each passage (*Figure 1C*), corresponding to a reading speed of 457 ± 142 words/min when divided by the number of words per passage. The speed was almost twice the normal reading speed for native readers (*Rayner, 1998*), indicating a specialized reading strategy for the task.

Next, we employed eye tracking to quantify how the readers allocated their attention to achieve effective reading and analyze which computational models could explain the reading time on each word, that is, the total fixation duration on each word during passage reading. In other words, we probed into what kind of computational principles could generate human-like attention distribution during goal-directed reading. A simple heuristic strategy was to attend to words that were orthographically or semantically similar to the words in the question (*Figure 1—figure supplement 1A*). The predictions of the heuristic models were not highly correlated with the human word reading time, and the predictive power, that is, the Pearson correlation coefficient between the predicted and real word reading time, was around 0.2 (*Figure 3—figure supplement 1A*).

The DNN models analyzed here, that is, SAR, BERT, ALBERT, and RoBERTa, all employed the attention mechanism to integrate over context to find optimal question answering strategies. Roughly speaking, the attention mechanism applied a weighted integration across all input words to generate

a passage-level representation and decide whether an option was correct or not, and the weight on each word was referred to as the attention weight (see *Figure 1—figure supplement 1B* and *Figure 2B* for illustrations about the attention mechanisms in the SAR and transformer-based models, respectively). When the attention weights of the SAR were used to predict the human word reading time, the predictive power was about 0.1 (*Figure 3A*, *Supplementary file 1a*).

In contrast to assigning a single weight on a word, the transformer-based model employed a multi-head attention mechanism: Each of the 12 layers had 12 parallel attention modules, that is, heads. Consequently, each word had 144 attention weights (12 layers × 12 heads), which were used to model the word reading time of humans based on linear regression. Since the attention weights of three transformer-based models showed comparable power to predict human word reading time, we reported the predictive power averaged over models (see *Figure 3—figure supplement 1A* for the results of individual models). The attention weights of randomly initialized transformer-based models could predict the human word reading time and the predictive power, which was around 0.3, was significantly higher than the chance level and the SAR (*Figure 3A*, *Supplementary file 1a*). The attention weights of pre-trained transformer-based models could also predict the human word reading time, the predictive power was around 0.5, significantly higher than the predictive power of heuristic models, the SAR, and randomly initialized transformer-based models (*Figure 3A*, *Supplementary file 1a*). The predictive power was further boosted for local but not global questions when the models were fine-tuned to perform the goal-directed reading task (*Figure 3A*, *Supplementary file 1a*). The weights assigned to attention heads in the linear regression are shown in *Figure 3—figure supplement 2*. For the fine-tuned models, we also predict the human word reading time using an unweighted averaged of the 144 attention heads and the predictive power was 0.3, significantly higher than that achieved by the attention weights of SAR (p=4 × 10⁻⁵, bootstrap). These results suggested that the human attention distribution was consistent with the attention weights in transformer-based models that were optimized to perform the same goal-directed reading task.

## Factors influencing human word reading time

The attention weights in transformer-based DNN models could predict the human word reading time. Nevertheless, it remained unclear whether such predictions were purely driven by basic text features that were known to modulate word reading time. Therefore, in the following, we first analyzed how basic text features modulated the word reading time during the goal-directed reading task, and then checked whether transformer-based DNNs could capture additional properties of the word reading time that could not be explained by basic text features.

Here, we further decomposed text features into visual layout features, that is, position of a word on the screen, and word features, for example, word length, frequency, and surprisal. Layout features were features that were mostly induced by line changes, which could be extracted without recognizing the words, while word features were finer-grained features that could only be extracted when the word or neighboring words were fixated. Linear regression analyses revealed layout features could significantly predict the word reading time (*Figure 3B*, *Supplementary file 1b*). Furthermore, the predictive power was higher for global than local questions (p=4 × 10⁻⁵, bootstrap, false discovery rate [FDR] corrected for comparisons across three features, i.e., layout features, word features, and question relevance), suggesting a question-type-specific reading strategy. Word features could also significantly predict human reading time, even when the influence of layout features was regressed out. Additionally, a linear mixed effect model revealed significant fixed effects for question type and all text/task-related features, as well as significant interactions between question type and these text/task-related features (*Supplementary file 1c*; *Pinheiro and Bates, 2006*; *Kuznetsova et al., 2017*).

The predictive power of the layout and word features, however, was lower than the predictive power of attention weights of transformer-based models (p=4 × 10⁻⁵, bootstrap, FDR corrected for comparisons across two features, i.e., layout and word features). When the layout and word features were regressed out, the residual word reading time was still significantly predicted by the attention weights in transformer-based models (*Figure 3—figure supplement 1B*, predictive power about 0.3). This result indicated that what the transformer-based models extracted were more than basic text features. Next, we analyzed whether the transformer-based models, as well as the human word reading time, were sensitive to task-related features. To characterize the relevance of each word to the question answering task, we asked another group of participants to annotate which words contributed most to

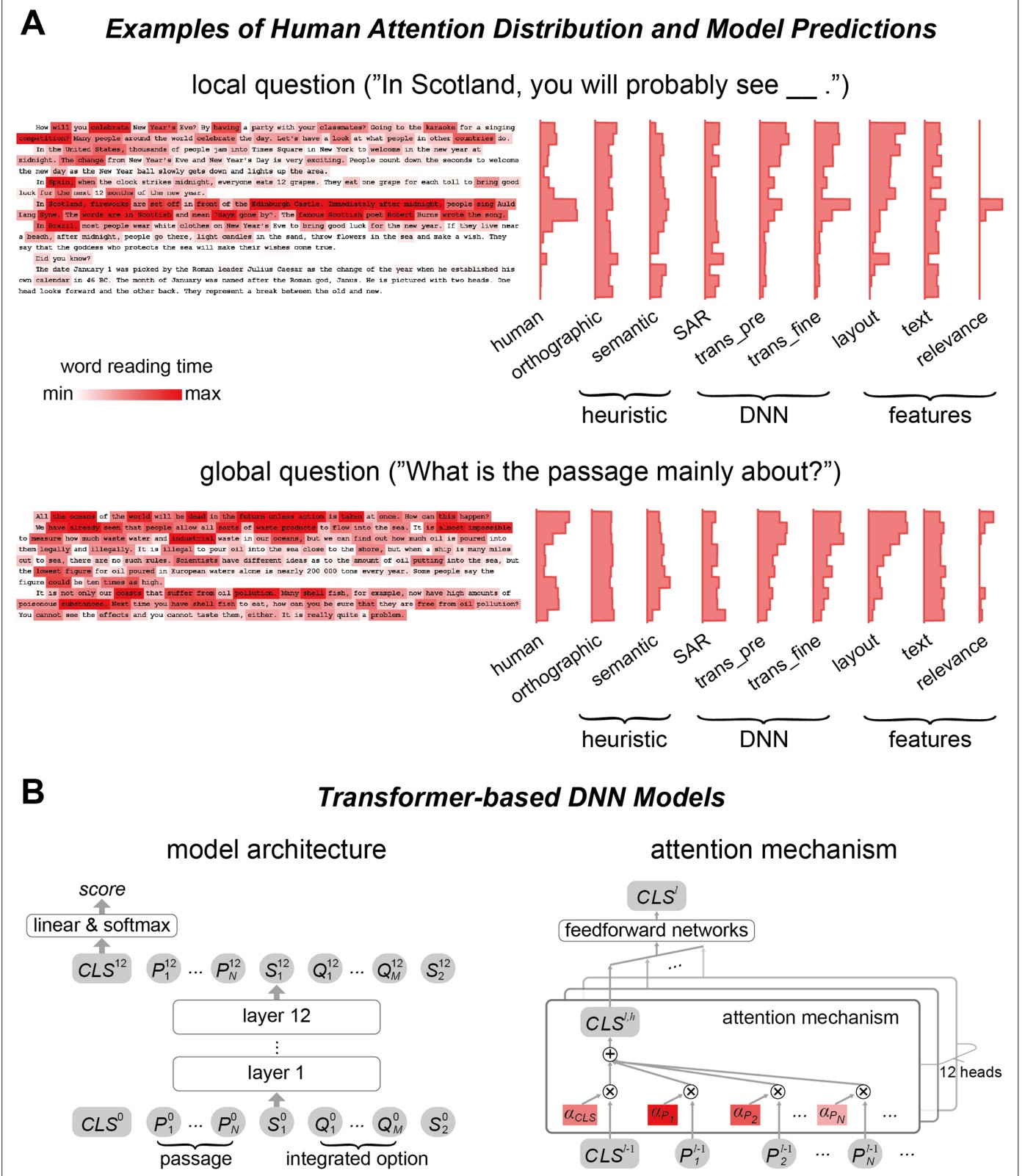

**Figure 2.** Human attention distribution and computational models. (**A**) Examples of human attention distribution, quantified by the word reading time. The histograms on the right showed the mean reading time on each line for both human data and model predictions. trans_pre: pre-trained transformer-based models; trans_fine: transformer-based models fine-tuned on the goal-directed reading task. (**B**) The general architecture of the 12-layer transformer-based models. The model input consists of all words in the passage and an integrated option. Output of the model relies on the

*Figure 2 continued on next page*

*Figure 2 continued*

node CLS (*Legge et al., 2002*), which is used to calculate a score reflecting how likely an option is the correct answer. The CLS node is a weighted sum of the vectorial representations of all words and tokens, and the attention weight for each word in the passage, that is, $\alpha$, is the deep neural network (DNN) attention analyzed in this study.

question answering. The annotated question relevance could significantly predict word reading time, even when the influences of layout and word features were regressed out (*Figure 3B*, *Supplementary file 1b*). When the question relevance was also regressed out, the residual word reading time was still significantly predicted by the attention weights in transformer-based models (*Figure 3—figure supplement 1C*, p=0.003, bootstrap, FDR corrected for comparisons across 12 models × 6 question types), but the predictive power dropped to about 0.2. Furthermore, a linear mixed effect model also revealed that more than 85% of the DNN attention heads contribute to the prediction of human reading time when considering text features and question relevance as covariates (*Supplementary file 1c*). These results demonstrated that the DNN attention weights provided additional information about the human word reading time than the text-related and task-related features analyzed here.

Further analyses revealed two properties of the distribution of question-relevant words. First, for local questions, the question-relevant words were roughly uniformly distributed in the passage, while for global questions, the question-relevant words tended to be near the passage beginning (*Figure 3—figure supplement 3A*). The eye-tracking data showed that readers also spent more time reading the passage beginning for global than local questions (*Figure 3C*), explaining why layout features more strongly influenced the answering of global than local questions. Second, few lines in the passage were question relevant (*Figure 3—figure supplement 3B*), and the eye-tracking data showed that readers spent more time reading the line with the highest question relevance (*Figure 3D*), confirming the influence of question relevance on word reading time.

## Attention in different processing stages for humans and DNNs

Next, we investigated whether humans and DNNs attended to different features in different processing stages. The early stage of human reading was indexed by the gaze duration, that is, duration of first-pass reading of a word, and the later stage was indexed by the counts of rereading. Results showed the influence of layout features increased from early to late reading stages for global but not local questions (*Figure 4A*, *Supplementary file 1d*). Consequently, the passage beginning effect differed between global and local questions only for the late reading stage (*Figure 4—figure supplement 1A*). The influence of word features did not strongly change between reading stages, while the influence of question relevance significantly increased from early to late reading stages (*Figure 4A*, *Figure 4—figure supplement 1B*). These results suggested that attention to basic text features developed early, while the influence of task mainly influenced late reading processes.

In the following, we further investigated whether transformer-based DNN attended to different features in different layers, which represented different processing stages. This analysis did not include layout features that were not available to the models. The attention weights in shallow layers were sensitive to word features in randomized, pre-trained, and fine-tuned models (*Figure 4B and C*). Only in the fine-tuned models, however, the attention weights in deep layers were sensitive to question relevance (see *Figure 4—figure supplements 2 and 3* for results of individual models). Therefore, the shallow and deep layers separately evolved text-based and goal-directed attention, and goal-directed attention was induced by fine-tuning on the task.

## Experiment 2: Question type specificity of the reading strategy

In Experiment 1, different types of questions were presented in blocks which encouraged the participants to develop question type-specific reading strategies. Next, we ran Experiment 2, in which questions from different types were mixed and presented in a randomized order, to test whether the participants developed question type-specific strategies in Experiment 1. Since it was time consuming to measure the response to all 800 questions, we randomly selected 96 questions for Experiment 2 (16 questions per type). In Experiment 2, the reading speed was on average 298 ± 123 words/min, lower than the speed in Experiment 1 (p=6 × 10$^{-4}$, bootstrap, FDR corrected for the comparisons across four experiments), but still much faster than normal reading speed (*Rayner, 1998*).

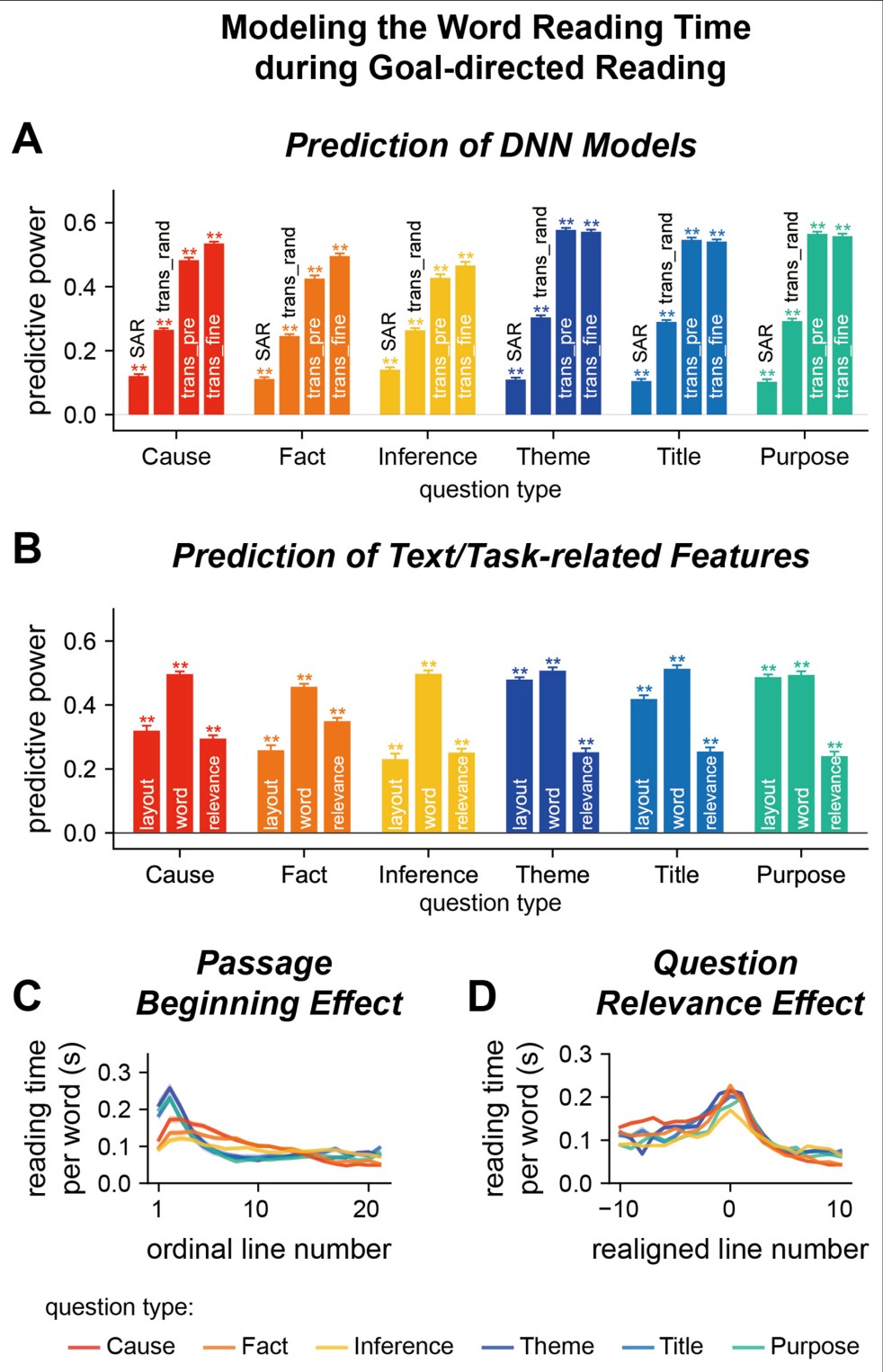

**Figure 3.** Model word reading time in Experiment 1. (**A, B**) Predict the word reading time based on the attention weights of deep neural network (DNN) models, text features, or question relevance. The predictive power is the correlation coefficient between the predicted word reading time and the actual word reading time. Predictive power significantly higher than chance is denoted by stars on the top of each bar. **p<0.01. trans_rand: transformer-base models with randomized parameters; trans_pre: pre-trained transformer-based models; trans_fine: transformer-based models fine-tuned on the goal-directed reading task. (**C**) Relationship between the word reading time and line index. The word reading time is longer near the beginning of a passage and the effect is

*Figure 3 continued on next page*

*Figure 3 continued*

stronger for global questions than local questions. (**D**) Relationship between the word reading time and question relevance. Line 0 refers to the line with the highest question relevance. The word reading time is higher for the question-relevant line. Color indicates the question type. The shade area indicates 1 standard error of the mean (SEM) across participants ($N$ = 25).

The online version of this article includes the following figure supplement(s) for figure 3:

**Figure supplement 1.** Transformer-based models can explain word reading time even when the influences of text features and question relevance are regressed out.

**Figure supplement 2.** Weights on individual attention heads in the linear regression when predicting human word reading time.

**Figure supplement 3.** Properties of the question relevance of words.

The word reading time was better predicted by fine-tuned than pre-trained transformer-based models (*Figure 5A*, *Supplementary file 1e*). For the influence of text and task-related features, compared to Experiment 1, the predictive power in Experiment 2 was higher for layout and word features, but lower for question relevance (*Figure 5B*, *Supplementary file 1f*). For local questions, consistent with Experiment 1, the effects of question relevance significantly increased from early to late processing stages that are separately indexed by gaze duration and counts of rereading (*Figure 5—figure supplement 1A*, *Supplementary file 1d*). The passage beginning effect was higher for global than local questions (*Figure 5C*, second column, p=$2 \times 10^{-4}$, bootstrap, FDR corrected for the comparisons across four experiments), but the difference was smaller than in Experiment 1 (*Figure 5C*, *Figure 5—figure supplement 2A*, p=$2 \times 10^{-4}$, bootstrap, FDR corrected for the comparisons across four experiments). The question relevance effect was also smaller in Experiment 2 than Experiment 1 (*Figure 5D*, *Figure 5—figure supplement 2B*, p=$2 \times 10^{-4}$, bootstrap, FDR corrected for the comparisons across four experiments). All these results indicated that the readers developed question type-specific strategies in Experiment 1, which led to faster reading speed and stronger task modulation of word reading time.

## Experiment 3: Effect of language proficiency

Experiments 1 and 2 recruited L2 readers. To investigate how language proficiency influenced task modulation of attention and the optimality of attention distribution, we ran Experiment 3, which was the same as Experiment 2 except that the participants were native English readers. In Experiment 3, the reading speed was on average 506 ± 155 words/min, higher than that in Experiment 2 (p=$6 \times 10^{-4}$, bootstrap, FDR corrected for the comparisons across four experiments). The question answering accuracy was comparable to L2 readers (*Figure 1B*).

The word reading time for native readers was slightly better predicted by fine-tuned than pre-trained transformer-based models (*Figure 5A*, *Supplementary file 1e*). For the influence of text and task-related features, compared to Experiment 2, the predictive power in Experiment 3 was higher for word features, but lower for layout features and question relevance (*Supplementary file 1f*). For local questions, the layout effect was more salient for gaze duration than for counts of rereading. In contrast, the effect of word-related features and task relevance was more salient for counts of rereading than gaze duration (*Figure 5—figure supplement 1B*, *Supplementary file 1d*). The passage beginning effect was higher for global than local questions, but the difference was smaller than in Experiment 2 (*Figure 5C*, *Figure 5—figure supplement 2A*, p = $2 \times 10^{-4}$, bootstrap, FDR corrected for the comparisons across four experiments). The question relevance effect was also smaller for Experiment 3 than Experiment 2 (*Figure 5D*, *Figure 5—figure supplement 2B*, p=$2 \times 10^{-4}$, bootstrap, FDR corrected for the comparisons across four experiments). These results showed that the word reading time of native readers was significantly modulated by the task, but the effect was weaker than that on L2 readers.

## Experiment 4: General-purpose reading

In the goal-directed reading task, participants read a passage to answer a question that they knew in advance, and the eye-tracking results revealed that participants spent more time reading question-relevant words. Question-relevant words, however, were generally longer content words (*Figure 3—figure supplement 3C and D*) that were often associated with longer reading time even without a

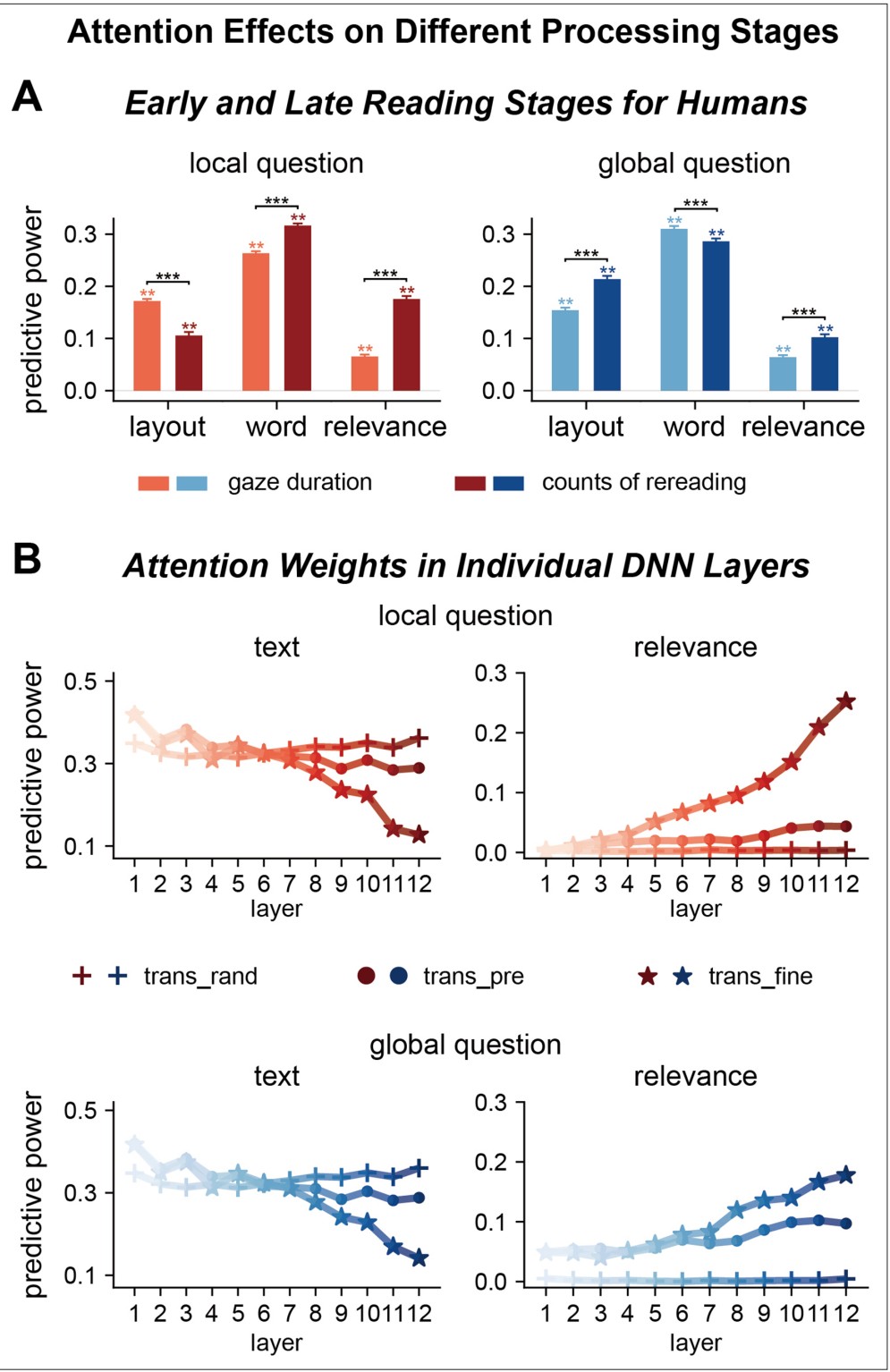

**Figure 4.** Factors influencing attention distribution in different processing stages for humans and deep neural networks (DNNs). (**A**) Human attention in early and late reading stages is differentially modulated by text features and question relevance. The early and late stages are separately characterized by gaze duration, that is, duration for the first reading of a word, and counts of rereading, respectively. **p<0.01; ***p<0.001. (**B**) DNN attention weights in different layers are also differentially modulated by text features and question relevance. Each attention head is separately modeled and averaged within each layer, and the results are further averaged across the three transformer-based models. Shallow layers of both fine-tuned and pre-trained models are more sensitive to text

*Figure 4 continued on next page*

*Figure 4 continued*

features. Deep layers of fine-tuned models are sensitive to question relevance. trans_rand: transformer-base models with randomized parameters; trans_pre: pre-trained transformer-based models; trans_fine: transformer-based models fine-tuned on the goal-directed reading task.

The online version of this article includes the following figure supplement(s) for figure 4:

**Figure supplement 1.** Passage beginning effects (**A**) and question relevance effects (**B**) in early and late reading stages.

**Figure supplement 2.** Factors influencing attention weights in each layer of deep neural networks (DNNs) for local questions.

**Figure supplement 3.** Factors influencing attention weights in each layer of deep neural networks (DNNs) for global questions.

task (*Rayner, 1998*). Therefore, to validate the question relevance effect, we ran Experiment 4 in which the participants read the passages without knowing the question to answer. The experiment used the same 96 questions as in Experiments 2 and 3, but adopted a different experimental procedure: participants previewed a passage before reading the question and were allowed to read the passage again to answer the question. We analyzed the reading pattern during passage preview, which was referred to as general-purpose reading.

The participants were given 1.5 min to preview the passage, and the reading speed was on average $225 \pm 40$ words/min, lower than that in Experiments 1–3 ($p=6 \times 10^{-4}$, bootstrap, FDR corrected for comparisons across four experiments). Before question answering, they were given another 0.5 min to reread the passage, but on average they spent only 0.04 min on rereading it. During passage preview, the word reading time was similarly predicted by the pre-trained and fine-tuned transformer-based models (*Figure 5A*, *Supplementary file 1e*). Furthermore, the word reading time was significantly predicted by layout and word features, but not question relevance (*Figure 5B*, *Supplementary file 1e*). Both the early and late processing stages of human reading were significantly affected by layout and word features, and the effects were larger for the late processing stage indexed by counts of rereading (*Figure 5—figure supplement 1C*, *Supplementary file 1d*). The passage beginning effect was not significantly different between local and global questions (*Figure 5C*, fourth column, $p=0.994$, bootstrap, FDR corrected for comparisons across four experiments), and the question relevance effect was significantly smaller than the question relevance effect in Experiments 1–3 (*Figure 5D*, *Figure 5— figure supplement 2B*, $p=2 \times 10^{-4}$, bootstrap, FDR corrected for comparisons across four experiments). These results confirmed that the question relevance effects observed during goal-directed reading were indeed task dependent.

## Discussion

Attention is a crucial mechanism to regulate information processing in the brain, and it has been hypothesized that a common computational role of attention is to optimize task performance. Previous support for the hypothesis mostly comes from tasks for which the optimal strategy can be easily derived. The current study, however, considers a real-world reading task in which the participants have to actively sample a passage to answer a question that cannot be answered by simple word-level orthographic or semantic matching. In this challenging task, it is demonstrated that human attention distribution can be explained by the attention weights in transformer-based DNN models that are optimized to perform the same reading task but blind to the human eye-tracking data. Furthermore, when participants scan a passage without knowing the question to answer, their attention distribution can also be explained by transformer-based DNN models that are optimized to predict a word based on the context.

Furthermore, we demonstrate that both humans and transformer-based DNN models achieve task-optimal attention distribution in multiple steps: For humans, basic text features strongly modulate the duration of the first reading of a word, while the question relevance of a word only modulates how many times the word is reread, especially for high-proficiency L2 readers compared to native readers. Similarly, the DNN models do not yield a single attention distribution, and instead they generate multiple attention distributions, that is, heads, for each layer. Here, we demonstrate that basic text features mainly modulate the attention weights in shallow layers, while the question relevance of

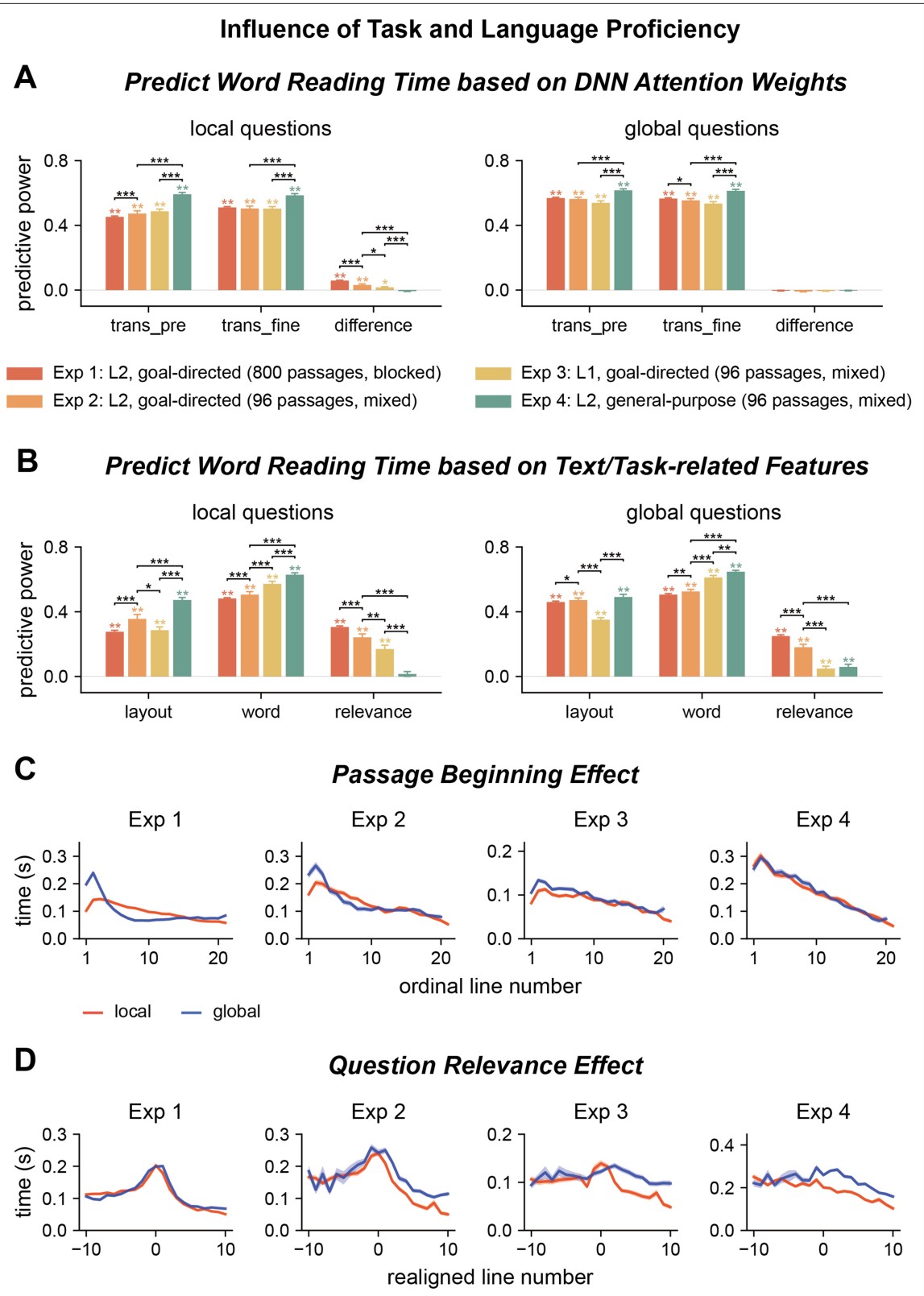

**Figure 5.** Influence of task and language proficiency on word reading time. (**A, B**) Predict the word reading time using attention weights of deep neural network (DNN) models, text features, and question relevance for all four experiments. Predictive power significantly higher than chance is marked by stars of the same color as the bar. Significant differences between experiments are denoted by black stars. trans_pre: pre-trained transformer-based

*Figure 5 continued on next page*

*Figure 5 continued*

models; trans_fine: transformer-based models fine-tuned on the goal-directed reading task. *p<0.05; **p<0.01; ***p<0.001. (**C, D**) Passage beginning and question relevance effects for all four experiments. The shade area indicates 1 SEM across participants (N = 25 for Exp 1; N = 20 for Exps 2-4).

The online version of this article includes the following figure supplement(s) for figure 5:

**Figure supplement 1.** Factors influencing human reading in different processing stages in Experiment 2 (**A**), Experiment 3 (**B**), and Experiment 4 (**C**).

**Figure supplement 2.** Passage beginning effects (**A**) and question relevance effects (**B**) in four experiments.

a word modulates the attention weights in deep layers, reflecting hierarchical control of attention to optimize task performance. The attention weights in both the shallow and deep layers of DNN contribute to the explanation of human word reading time (*Figure 3—figure supplement 2*).

## Computational models of attention

A large number of computational models of attention have been proposed. According to Marr's three levels of analysis (*Marr, 1982*), some models investigate the computational goal of attention (*Dayan et al., 2000*; *Legge et al., 2002*) and some models provide an algorithmic implementation of how different factors modulate attention (*Reichle et al., 2003*; *Itti et al., 1998*). Computationally, it has been hypothesized that attention can be interpreted as a mechanism to optimize learning and decision-making, and empirical evidence has been provided that the brain allocates attention among different information sources to optimally reduce the uncertainty of a decision (*Dayan et al., 2000*; *Gottlieb et al., 2014*; *Legge et al., 2002*). The current study provides critical support to this hypothesis in a real-world task that engages multiple forms of attention, for example, attention to visual layout features, attention to word features, and attention to question-relevant information. These different forms of attention, which separately modulate different eye-tracking measures (*Figure 4A*), jointly achieve an attention distribution that is optimal for question answering.

The transformer-based DNN models analyzed here are optimized in two steps, that is, pre-training and fine-tuning. The results show that pre-training leads to text-based attention that can well explain general-purpose reading in Experiment 4, while the fine-tuning process leads to goal-directed attention in Experiments 1–3 (*Figures 4B and 5A*). Pre-training is also achieved through task optimization, and the pre-training task used in all the three models analyzed here is to predict a word based on the context. The purpose of the word prediction task is to let models learn the general statistical regularity in a language based on large corpora, which is crucial for model performance on downstream tasks (*Lan et al., 2020*; *Liu et al., 2019*; *Devlin et al., 2019*), and this process can naturally introduce the sensitivity to word surprisal, that is, how unpredictable a word is given the context. Previous eye-tracking studies have suggested that the predictability of words, that is, surprisal, can modulate reading time (*Hale, 2016*), and neuroscientific studies have also indicated that the cortical responses to language converge with the representations in pre-trained DNN models (*Goldstein et al., 2022*; *Schrimpf et al., 2021*). The results here further demonstrate that the DNN optimized for the word prediction task can evolve attention properties consistent with the human reading process. Additionally, the tokenization process in DNN can also contribute to the similarity between human and DNN attention distributions: DNN first separates words into tokens (e.g., 'tokenization' is separated into 'token' and 'ization'). Tokens are units that are learned based on co-occurrence of letters and is not strictly linked to any linguistically defined units. Since longer words tend to be separated into more tokens, that is, fragments of frequently co-occurred letters, longer words receive more attention even if the model pay uniform attention to each of its input, that is, a token.

A separate class of models investigates which factors shape human attention distribution. A large number of models are proposed to predict bottom-up visual saliency (*Tatler et al., 2011*; *Borji et al., 2013*), and recently DNN models are also employed to model top-down visual attention. It is shown that, through either implicit (*Anderson et al., 2018*; *Xu et al., 2015*) or explicit training (*Das et al., 2017*), DNNs can predict which parts of a picture relate to a verbal phrase, a task similar to goal-directed visual search (*Wolfe and Horowitz, 2017*). The current study distinguishes from these studies in that the DNN model is not trained to predict human attention. Instead, the DNN models naturally generate human-like attention distribution when trained to perform the same task that humans perform, suggesting that task optimization is a potential cause for human attention distribution during reading.

## Models for human reading and human attention to question-relevant information

How human readers allocate attention during reading is an extensively studied topic, mostly based on studies that instruct readers to read a sentence in a normal manner, not aimed to extract a specific kind of information (*Clifton et al., 2016*). Previous eye-tracking studies have shown that the readers fixate longer upon, for example, longer words, words of lower-frequency, words that are less predictable based on the context, and words at the beginning of a line (*Rayner, 1998*). A number of models, for example, the E-Z reader (*Reichle et al., 2003*) and SWIFT (*Engbert et al., 2005*), have been proposed to predict the eye movements during reading based on basic oculomotor properties or lexical processing (*Reichle et al., 2003*). Some models also view reading as an optimization process that minimizes the time or the number of saccades required to read a sentence (*Legge et al., 2002*; *Liu and Reichle, 2010*). These models can generate fine-grained predictions, for example, which letter in a word will be fixated first, for the reading of simple sentences, but have only been occasionally tested for complex sentences or multiline texts (*Mancheva et al., 2015*) or to characterize different reading tasks, for example, z-string reading and visual searching (*Reichle et al., 2012*).

When readers read a passage to answer a question that can be answered using a word-matching strategy (*Hermann et al., 2015*), a recent study has demonstrated that the specific reading goal modulates the word reading time and the effect can be modeled using an RNN model (*Hahn and Keller, 2023*). Here, we focus on questions that cannot be answered using a word-matching strategy (*Figure 1B*) and demonstrate that, for these challenging questions, attention is still modulated by the reading goal but the attention modulation cannot be explained by a word-matching model (*Figure 3—figure supplement 1*). Instead, the attention effect is better captured by transformer models than an advanced RNN model, that is, the SAR (*Figure 3A*). Combining the current study and the study by *Hahn and Keller, 2023*, it is possible that the word reading time during a general-purpose reading task can be explained by a word prediction task, the word reading time during a simple goal-directed reading task that can be solved by word matching can be modeled by an RNN model, while the word reading time during a more complex goal-directed reading task involving inference is better modeled using a transformer model. The current study also further demonstrates that elongated reading time on task-relevant words is caused by counts of rereading and further studies are required to establish whether earlier eye movement measures can be modulated by, for example, a word-matching task. In addition, future studies can potentially integrate classic eye movement models with DNNs to explain the dynamic eye movement trajectory, possibly with a letter-based spatial resolution.

When human readers read a passage with a particular goal or perspective, previous studies have revealed inconsistent results about whether the readers spent more time reading task-relevant sentences (*Yeari et al., 2015*; *Grabe, 1979*; *Kaakinen et al., 2002*). To explain the inconsistent results, it has been proposed that the question relevance effect weakens for readers with a higher working memory and when readers read a familiar topic (*Kaakinen et al., 2003*). Similarly, here, we demonstrate that non-native readers indeed spend more time reading question-relevant information than native readers do (*Figure 5D*, *Figure 5—figure supplement 2B*). Therefore, it is possible that when readers are more skilled and when the passage is relatively easy to read, their processing is so efficient so that they do not need extra time to encode task-relevant information and may rely on covert attention to prioritize the processing of task-relevant information.

## DNN attention to question-relevant information

A number of studies have investigated whether the DNN attention weights are interpretable, but the conclusions are mixed: some studies find that the DNN attention weights are positively correlated with the importance of each word (*Yang et al., 2016*; *Lin et al., 2017*), while other studies fail to find such correlation (*Serrano and Smith, 2019*; *Jain and Wallace, 2019*). The inconsistent results are potentially caused by the lack of gold standard to evaluate the contribution of each word to a task. A few recent studies have used the human word reading time as the criterion to quantify word importance, but these studies do not reach consistent conclusions either. Some studies find that the attention weights in the last layer of transformer-based DNN models better correlates with human word reading time than basic word frequency measures (*Bolotova et al., 2020*), and integrating human word reading time into DNN can slightly improve task performance (*Malmaud et al., 2020*). Other

studies, however, find no meaningful correlation between the attention weights in transformer-based DNNs and human word reading time (*Sood et al., 2020*).

The current results provide a potential explanation for the discrepancy in the literature: The last layer of transformer-based DNNs is tuned to task relevant information (*Figure 4B*), but the influence of task relevance on word reading time is rather weak for native readers (*Figure 5B*). Consequently, the correlation between the last-layer DNN attention weights and human reading time may not be robust. The current results demonstrate that the reading time of both native and non-native readers are reliably modulated by basic text features, which can be modeled by the attention weights in shallower DNN layers.

Finally, the current study demonstrates that transformer-based DNN models can automatically generate human-like attention in the absence of any prior knowledge about the properties of the human reading process. Simpler models that fail to explain human performance also fail to predict human attention distribution. It remains possible, however, different models can solve the same computational problem using distinct algorithms, and only some algorithms generate human-like attention distribution. In other words, human-like attention distribution may not be a unique solution to optimize the goal-directed reading task. Sharing similar attention distribution with humans, however, provides a way to interpret the attention weights in computational models. From this perspective, the dataset and methods developed here provide an effective probe to test the biological plausibility of NLP models that can be easily applied to test whether a model evolves human-like attention distribution.

## Materials and methods

### Participants

Totally, 162 participants took part in this study (19–30 years old, mean age, 22.5 years; 84 females). All participants had normal or corrected-to-normal vision. Experiment 1 had 102 participants. Experiments 2–4 had 20 participants. No participant took part in more than one experiment. Additional 17 participants were recruited but failed to pass the calibration process for eye tracking and therefore did not participant in the reading experiments.

In Experiments 1, 2, and 4, participants were native Chinese readers. They were college students or graduate students from Zhejiang University, and were thus above the level required to answer high-school-level reading comprehension questions. English proficiency levels were further guaranteed by the following criterion for screening participants: a minimum score of 6 on IELTS, 80 on TOEFL, or 425 on CET6 (The National College English Test (CET) is a national English test system developed to examine the English proficiency of college students in China. The CET includes tests of two levels: a lower level test CET4 and a higher level test CET6.). In Experiment 3, participants were native English readers. The experimental procedures were approved by the Research Ethics Committee of the College of Medicine, Zhejiang University (2019-047). The participants provided written consent and were paid.

### Experimental materials

The reading materials were selected and adapted from the large-scale RACE dataset, a collection of reading comprehension questions in English exams for middle and high schools in China (*Lai et al., 2017*). We selected 800 high-school-level questions from the test set of RACE and each question was associated with a distinct passage (117–456 words per passage). All questions were multiple-choice questions with four alternatives including only one correct option among them. The questions fell into six types, that is, Cause (N = 200), Fact (N = 200), Inference (N = 120), Theme (N = 100), Title (N = 100), and Purpose (N = 80). The Cause, Fact, and Inference questions concerned the location, extraction, and comprehension of specific information from a passage, and were referred to as local questions. Questions of Theme, Title, and Purpose tested the understanding of a passage as a whole and were referred to as global questions.

In a separate online experiment, we acquired annotations about the relevance of each word to the question answering task. For each passage, a participant was allowed to annotate up to five key words that were considered relevant to answering the corresponding question. Each passage was annotated by $N$ participants ($N \geq 26$), producing $N$ versions of annotated key words. Each version of annotation

was then validated by a separate participant. In the validation procedure, the participant was required to answer the question solely based on the key words of a specific annotation version; if the person could not derive the correct answer, this version of annotation was discarded. The percentage of questions correctly answered in the validation procedure was 75.9 and 67.6% for local and global questions, respectively. If $M$ versions of annotation passed the validation procedure and a word was annotated in $K$ versions, the question relevance of the word was $K/M$. More details about the question types and the annotation procedures could be found in *Zou et al., 2021*.

## Experimental procedures

### Experiment 1

Experiment 1 included all 800 passages, and different question types were separately tested in different sessions, hence six sessions in total. Each session included 25 participants and one participant could participate in multiple sessions. Before each session, participants were familiarized with five questions that were not used in the formal session. During the formal session, questions were presented in a randomized order. Considering the quantities of questions, for Cause and Fact questions, the session was carried out in three separate days (one third questions on each day), and for other question types, the session was carried out in two separate days (50% of questions on each day).

The experiment procedure in Experiment 1 is illustrated in *Figure 1A*. In each trial, participants first read a question, pressed the space bar to read the corresponding passage, pressed the space bar again to read the question coupled with four options, and chose the correct answer. The time limit for passage reading was 120 s. To encourage the participants to read as quickly as possible, the bonus they received for a specific question would decrease linearly from 1.5 to 0.5 RMB over time. They did not receive any bonus for the question, however, if they gave a wrong answer. Furthermore, before answering the comprehension question, the participants reported whether they were confident about that they could correctly answer the question (yes or no). Participants selected yes for 90.47% of questions (89.62 and 92.04% for local and global questions, respectively). After answering the question, they also rated their confidence about their answer on the scale of 1–4 (low to high). The mean confidence rating was 3.25 (3.28 and 3.18 for local and global question, respectively), suggesting that the participants were confident about their answers.

### Experiments 2 and 3

Experiments 2 and 3 included 96 reading passages and questions that were randomly selected from the questions used in Experiment 1 and included 16 questions for each question type. The six types of questions were mixed and presented in a randomized order. The trial structure, as well as the familiarization procedure, in Experiments 2 and 3 was identical to that in Experiment 1. Experiments 2 and 3 were identical except that Experiment 2 recruited high-proficiency L2 readers while Experiment 3 recruited native English readers.

### Experiment 4

Experiment 4 included the 96 questions presented in Experiments 2 and 3, which were presented in a randomized order. The trial structure in Experiment 4 is similar to that in Experiments 1–3, except that a 90 s passage preview stage was introduced at the beginning of each trial. During passage preview, participants had no prior information of the relevant question. The participants could press the space bar to terminate the preview and to read a question. Then, participants read the passage again with a time limit of 30 s, before proceeding to answer the question. The payment method was similar to Experiment 2, and the bonus was calculated based on the duration of second-pass passage reading.

## Stimulus presentation and eye tracking

The text was presented using the bold Courier New font, and each letter occupied 14 × 27 pixels. We set the maximum number of letters on each line to 120 and used double space. We separated paragraphs by indenting the first line of each new paragraph. Participants sat about 880 mm from a monitor, at which each letter horizontally subtended approximately 0.25° of visual angle.

Eye-tracking data were recorded from the left eye with 500 Hz sampling rate (Eyelink Portable Duo, SR Research). The experiment stimuli were presented on a 24-inch monitor (1920 × 1080 resolution; 60 Hz refresh rate) and administered using MATLAB Psychtoolbox (*Brainard, 1997*). Each experiment

started with a 13-point calibration and validation of eye tracker, and the validation error was required to be below 0.5° of visual angle. Furthermore, before each trial, a 1-point validation was applied, and if the calibration error was higher than 0.5° of visual angle, a recalibration was carried out. Head movements were minimized using a chin and forehead rest.

## Word-level reading comprehension models

The orthographic and semantic models probed whether the reading comprehension questions could be answered based on word-level orthographic or semantic information. Both models calculated the similarity between each content word in the passage and each content word in an option, and averaged the word-by-word similarity across all words in the passage and all words in the option (*Figure 1—figure supplement 1A*). The option with the highest mean similarity value was chosen as the answer. For the orthographic model, similarity was quantified using the edit distance (*Levenshtein, 1966*). For the semantic model, similarity was quantified by the correlation between vectorial representations of word meaning, that is, the glove model (*Pennington et al., 2014*). Performance of the models remained similar if the answer was chosen based on the maximal word-by-word similarity, instead of the mean similarity.

## RNN-based reading comprehension models

The SAR was a classical RNN-based model for the reading comprehension task (*Chen et al., 2016*). In contrast to the word-level models, the SAR was context sensitive and employed bidirectional RNNs to integrate information across words (*Figure 1—figure supplement 1B*). Independent bidirectional RNNs were employed to build a vectorial representation for the question and each option. An additional bidirectional RNN was applied to construct a vectorial representation for each word in the passage, and a passage representation was built by a weighted sum of the representations of individual words in the passage. The weight on each word, that is, the attention weight, captured the similarity between the representation of the word and the question representation using a bilinear function. Finally, based on the passage representation and each option representation, a bilinear dot layer calculated the possibility that the option was the correct answer.

## Transformer-based reading comprehension models

We tested three popular transformer-based DNN models, that is, BERT (*Devlin et al., 2019*), ALBERT (*Lan et al., 2020*), and RoBERTa (*Liu et al., 2019*), which were all reported to reach high performance on the reading comprehension task. ALBERT and RoBERTa were both adapted from BERT and had the same basic structure. RoBERTa differed from BERT in its pre-training procedure (*Liu et al., 2019*) while ALBERT applied factorized embedding parameterization and cross-layer parameter sharing to reduce memory consumption (*Lan et al., 2020*). Following previous studies (*Lan et al., 2020*; *Liu et al., 2019*), each option was independently processed. For the $i$th option ($i$ = 1, 2, 3, or 4), the question and the option were concatenated to form an integrated option. As shown in the left panel of *Figure 2B*, for the $i$th option, the input to models was the following sequence:

$$CLS_i, P_1, P_2, ..., P_N, S_{i,1}, O_{i,1}, O_{i,2}, ..., O_{i,M}, S_{i,2},$$

where $CLS_i$, $S_{i,1}$, and $S_{i,2}$ denote special tokens separating different components of the input. $P_1$, $P_2$, ..., $P_N$ denote all the $N$ words of a passage, and $O_{i,1}$, $O_{i,2}$, ..., $O_{i,M}$ denote all the $M$ words in the $i$th integrated option. Each of the token was represented by a vector. The vectorial representation was updated in each layer, and in the following the output of the $l$th layer is denoted as a superscript, for example, $CLS_i^l$. Following previous studies (*Lan et al., 2020*; *Liu et al., 2019*), we calculated a score for each option, which indicated the possibility that the option was the correct answer. The score was calculated by first applying a linear transform to the final representation of the CLS token, that is,

$$s_i = \Phi CLS_i^{12},$$

where $CLS_i^{12}$ is the final output representation of CLS and $\Phi$ is a vector learned from data. The score was independently calculated for each option and then normalized using the following equation:

$$score_i = \frac{exp\left(s_i\right)}{\sum_{i=1}^{4} exp\left(s_i\right)}$$

The answer to a question was determined as the option with the highest score, and all the models were trained to maximize the logarithmic score of the correct option. The transformer-based models were trained in two steps (*Figure 1D*). The pre-training process aimed to learn general statistical regularities in a language based on large corpora, that is, BooksCorpus (*Zhu et al., 2015*) and English Wikipedia, while the fine-tuning process trained models to perform the reading comprehension task based on RACE dataset. All models were implemented based on HuggingFace (*Wolf et al., 2020*), and all hyperparameters for fine-tuning were adopted from previous studies (*Lan et al., 2020*; *Liu et al., 2019*; *Zhang et al., 2020*; *Ran et al., 2019*; see *Supplementary file 1g*).

## Attention in transformer-based models

The transformer-based models we applied had 12 layers, and each layer had 12 parallel attention heads. Each attention head calculated an attention weight between any pair of inputs, including words and special tokens. The vectorial representation of each input was then updated by the weighted sum of the vectorial representations of all inputs (*Vaswani et al., 2017*). Since only the CLS token was directly related to question answering, here we restrained the analysis to the attention weights that were used to calculate the vectorial representation of CLS (*Figure 2B*, right panel). In the $h$th head, the vectorial representation of CLS was computed using the following equations. For the sake of clarity, we did not distinguish the input words and special tokens and simply denoted them as $X_i$.

$$CLS^h = \sum_{i=1}^{N+M+3} \alpha_i V_i = \alpha_{CLS} V_{CLS} + \sum_{n=1}^{N} \alpha_{Pn} V_{Pn} + \alpha_{S1} V_{S1} + \sum_{m=1}^{M} \alpha_{Om} V_{Om} + \alpha_{S2} V_{S2},$$

$$\alpha_i = \frac{exp\left(Q_{CLS} K_i^{\mathrm{T}}\right)}{\sum_{i=1}^{N+M+3} exp\left(Q_{CLS} K_i^{\mathrm{T}}\right)},$$

$$V_i = X_i W^V + b^V, \ K_i = X_i W^K + b^K, \ Q_{CLS} = X_{CLS} W^Q + b^Q,$$

where $W^V$, $W^Q$, $W^K$, $b^V$, $b^Q$, and $b^K$ are parameters to learn from the data, and $\alpha_i$ is the attention weight between CLS and $X_i$. The attention weight between CLS and the $n$th word in the passage, that is, $\alpha_{Pn}$, was compared to human attention. Here, we only considered the attention weights associated with the correct option. Additionally, DNNs used byte-pair tokenization which split some words into multiple tokens. We converted the token-level attention weights to word-level attention weights by summing the attention weights over tokens within a word (*Bolotova et al., 2020*; *Clark et al., 2019*).

## Eye-tracking measures

We analyzed eye movements during passage reading in Experiments 1–3, and the passage preview in Experiment 4. For each word, the total fixation time, gaze duration, and run counts was extracted using the SR Research Data Viewer software. The total fixation time of a word is referred to as the word reading time. The gaze duration was how long a word was fixated before the gaze moved to other words, reflected first-pass processing of a word. To characterize late processing of a word, we further calculated the counts of rereading, which were defined as the run counts minus 1. Words that were not reread were excluded from the analysis of counts of rereading. Each of the eye-tracking measure was averaged across all participants who correctly answered the question.

## Regression models

We employed linear regression to analyze how well each model, as well as each set of text/task-related features, could explain human attention measured by eye tracking. In all regression analyses, each regressor and the eye-tracking measure were normalized within each passage by taking the z-score. The predictive power, that is, the Pearson correlation coefficient between the predicted eye-tracking measure and the actual eye-tracking measure, was calculated based on fivefold cross-validation.

For the SAR, each word had one attention weight, which was used as the regressor. For transformer-based models, since each model contained 12 layers and each layer contained 12 attention heads, altogether there were 144 regressors. Text features included layout features and word features. The layout features concerned the visual position of text, including the coordinate of the leftmost pixel of a word, ordinal paragraph number of a word in a passage, ordinal line number of a word in a paragraph, and ordinal line number of a word in a passage. The word features included word length, logarithmic

word frequency estimated based on the BookCorpus (*Zhu et al., 2015*) and English Wikipedia using SRILM (*Stolcke, 2002*), and word surprisal estimated from GPT-2 Medium (*Radford et al., 2018*). The task-related feature referred to the question relevance annotated by another group of participants (see 'Experimental materials' for details).

Additionally, we also applied linear regression to probe how DNN attention was affected by text features and question relevance. Since information of lines and paragraphs was not available to DNNs, the layout features only included the ordinal position of a word in a sentence, ordinal position of a word in a passage, and ordinal sentence number of a word in this analysis.

## Linear mixed effect model

To characterize the influences of different factors on human word reading time, we employed linear mixed effects models (*Pinheiro and Bates, 2006*) implemented in the lmerTest package (*Kuznetsova et al., 2017*) of R. For the baseline model, we treated the type of questions (local vs. global; local = baseline) and all text/task-related features as fixed factors, and considered the interaction between the type of questions and these text/task-related features. We included participants and items (i.e., questions) as random factors, each with associated random intercepts. The formulation of the baseline model was *reading-time ~ ParagraphNumber * QuestionType + LineNumberInPassage * Question-Type + LeftMostPixel * QuestionType + LineNumberInParagraph * QuestionType + LogWordFreq * QuestionType + WordLength * QuestionType + Surprisal * QuestionType + QuestionRelevance * QuestionType + (1 | Participant) + (1 | question)*. Additionally, starting from the baseline model, we augmented the baseline model by adding DNN attention as additional fixed factors. This augmentation facilitated an examination of whether DNN attention demonstrated a statistically significant contribution to the prediction of human word reading time. Notably, the DNN attention was derived from diverse sources, including SAR, randomized BERT, pre-trained BERT, and fine-tuned BERT.

## Statistical tests

In the regression analysis, we employed a one-sided permutation test to test whether a set of features could statistically significantly predict an eye- tracking measure. A total of 500 chance-level predictive power was calculated by predicting the eye-tracking measure shuffled across all words within a passage: the eye-tracking measure to predict was shuffled but the features were not. The procedure was repeated 500 times, creating 500 chance-level predictive power. If the actual correlation was smaller than $N$ out of the 500 chance-level correlation, the significance level was (N + 1)/501.

When comparing the responses to local and global questions, the three types of local/global questions were pooled. The comparison between local and global questions, as well as the comparison between experiments, was based on bias-corrected and accelerated bootstrap (*Efron and Tibshirani, 1994*). For example, to test whether the predictive power differed between the two types of questions, all global questions were resampled with replacement 50,000 times and each time the predictive power was calculated based on the resampled questions, resulting in 50,000 resampled predictive power. If the predictive power for local questions was greater (or smaller) than $N$ out of the 50,000 resampled predictive power for global questions, the significance level of their difference was 2(N + 1)/50,001. When multiple comparisons were performed, the p-value was further adjusted using the FDR correction.

## Acknowledgements

We thank David Poeppel, Yunyi Pan, and Erik D Reichle for valuable comments on earlier versions of this manuscript; Jonathan Simon, Bingjiang Lyu, and members of the Ding lab for thoughtful discussions and feedback; Qian Chu, Yuhan Lu, Anqi Dai, Zhonghua Tang, and Yan Chen for assistance with experiments. This work was supported by STI2030-Major Project 2021ZD0204105, National Natural Science Foundation of China 32222035, National Natural Science Foundation of China 32300856, Major Scientific Research Project of Zhejiang Lab 2019KB0AC02, and Fundamental Research Funds for the Central Universities 226-2023-00091.

## Additional information

### Funding

| Funder | Grant reference number | Author |
|---|---|---|
| STI2030-Major Project | 2021ZD0204105 | Nai Ding |
| National Natural Science Foundation of China | 32222035 | Nai Ding |
| National Natural Science Foundation of China | 32300856 | Jiajie Zou |
| Major Scientific Project of Zhejiang Laboratory | 2019KB0AC02 | Nai Ding |
| Fundamental Research Funds for the Central Universities | 226-2023-00091 | Nai Ding |

The funders had no role in study design, data collection and interpretation, or the decision to submit the work for publication.

### Author contributions

Jiajie Zou, Conceptualization, Data curation, Formal analysis, Investigation, Visualization, Methodology, Writing – original draft, Writing – review and editing; Yuran Zhang, Jialu Li, Data curation; Xing Tian, Supervision, Project administration; Nai Ding, Conceptualization, Formal analysis, Supervision, Funding acquisition, Writing – original draft, Project administration, Writing – review and editing

### Author ORCIDs

Jiajie Zou ⓘ http://orcid.org/0009-0009-9054-117X
Xing Tian ⓘ http://orcid.org/0000-0003-1629-6304
Nai Ding ⓘ http://orcid.org/0000-0003-3428-2723

### Ethics

Human subjects: The experimental procedures were approved by the Research Ethics Committee of the College of Medicine, Zhejiang University (2019-047). The participants provided written consent and were paid.

Reviewer #1 (Public Review): https://doi.org/10.7554/eLife.87197.3.sa1
Reviewer #2 (Public Review): https://doi.org/10.7554/eLife.87197.3.sa2
Reviewer #3 (Public Review): https://doi.org/10.7554/eLife.87197.3.sa3
Author Response https://doi.org/10.7554/eLife.87197.3.sa4

## Additional files

### Supplementary files

• Supplementary file 1. Supplementary tables. (a) p-values for the model prediction of word reading time. trans_rand: transformer-base models with randomized parameters; trans_pre: pre-trained transformer-based models; trans_fine: transformer-based models fine-tuned on the goal-directed reading task. (b) p-values for the prediction of word reading time using text or task-related features. (c) Linear mixed effects modeling of human word reading time. The question type is coded as 0 (local question) or 1 (global question), and other factors are continuous regressors. Given the substantial number of attention weights in BERT (i.e., 144), we present the 1st quartile and 3rd quartile values for b, SE, and t and report the ratio of attention weights that reach significant level. *b*: regression coefficient; *SE*: standard error of regression coefficient. (d) p-values for the prediction of early and late eye-tracking measures using text or task-related features. GD: gaze duration; CR: counts of rereading. (e) p-values for the prediction of word reading time for all four experiments. trans_pre: pre-trained transformer-based models; trans_fine: transformer-based models fine-tuned on the goal-directed reading task. (f) p-values for the comparisons between experiments. trans_pre: pre-trained transformer-based models; trans_fine: transformer-based models fine-tuned

on the goal-directed reading task. (g) Hyperparameters for DNN fine-tuning. We adapted these hyperparameters from references (*Lan et al., 2020*; *Liu et al., 2019*; *Zhu et al., 2015*; *Wolf et al., 2020*).

• MDAR checklist

### Data availability

All eye tracking data and code are available at (https://github.com/jiajiezou/TOA copy archived at *jiajiezou, 2023*).

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
