## [Editor Report · eLife assessment]

This study provides a **valuable** contribution to the study of eye-movements in reading, revealing that attention-weights from a deep neural network show a statistically reliable fit to the word-level reading patterns of humans. Its evidence is **convincing** and strengthens a line of research arguing that attention in reading reflects task optimization. The work would be of interest to psychologists, neuroscientists, and machine learning researchers.

---

## [Referee Report · Reviewer #1 (Public Review)]

This manuscript describes a set of four passage-reading experiments which are paired with computational modeling to evaluate how task-optimization might modulate attention during reading. Broadly, participants show faster reading and modulated eye-movement patterns of short passages when given a preview of a question they will be asked. The attention weights of a Transformer-based neural network (BERT and variants) show a statistically reliable fit to these reading patterns above-and-beyond text- and semantic-similarity baseline metrics, as well as a recurrent-network-based baseline. Reading strategies are modulated when questions are not previewed, and when participants are L1 versus L2 readers, and these patterns are also statistically tracked by the same transformer-based network.

Strengths:

- Task-optimization is a key notion in current models of reading and the current effort provides a computationally rigorous account of how such task effects might be modeled

- Multiple experiments provide reasonable effort towards generalization across readers and different reading scenarios

- Use of RNN-based baseline, text-based features, and semantic features provides a useful baseline for comparing Transformer-based models like BERT

Weaknesses:

- Generalization across neural network models may be limited (models differ in size, training data etc.); it is thus not always clear which specific model characteristics support their fit to human reading patterns.

---

## [Referee Report · Reviewer #2 (Public Review)]

In this study, researchers aim to understand the computational principles behind attention allocation in goal-directed reading tasks. They explore how deep neural networks (DNNs) optimized for reading tasks can predict reading time and attention distribution. The findings show that attention weights in transformer-based DNNs predict reading time for each word. Eye tracking reveals that readers focus on basic text features and question-relevant information during initial reading and rereading, respectively. Attention weights in shallow and deep DNN layers are separately influenced by text features and question relevance. Additionally, when readers read without a specific question in mind, DNNs optimized for word prediction tasks can predict their reading time. Based on these findings, the authors suggests that attention in real-world reading can be understood as a result of task optimization.

Strengths of the Methods and Results:

The present study employed stimuli consisting of paragraphs read by middle and high school students, covering a wide range of diverse topics. This choice ensured that the reading experience for participants remained natural, ultimately enhancing the ecological validity of the findings and conclusions.

In Experiments 1-3, participants were instructed to read questions before the text, while in Experiment 4 participants were instructed to read questions after the text. This deliberate manipulation allowed the paper to assess how different reading task conditions influence reading and eye movements.

Weaknesses of the Methods and Results:

While the study benefits from several strengths, it is important to acknowledge its limitations. Notably, recent months have seen significant advancements in Deep Neural Network (DNN) models, including the development of models such as GPT-3.5 and GPT-4, which have demonstrated remarkable capabilities in tasks resembling human cognition, like Theory of Mind. However, as the code for these cutting-edge models was not publicly accessible, they were unable to evaluate whether the attention mechanisms in the most up-to-date DNN models could provide improved predictions for human eye-movement data. This constraint represents a limitation in the investigation.

The methods and data presented in this study are valuable for gaining insights into the psychological mechanisms of reading. Moreover, the data provided in this paper may prove instrumental in enhancing the performance of future DNN models.

---

## [Referee Report · Reviewer #3 (Public Review)]

This paper presents several eyetracking experiments measuring task-directed reading behavior where subjects read texts and answered questions. It then models the measured reading times using attention patterns derived from deep-neural network models from the natural language processing literature. Results are taken to support the theoretical claim that human reading reflects task-optimized attention allocation.

Strengths:

(1) The paper leverages modern machine learning to model a high-level behavioral task (reading comprehension). While the claim that human attention reflects optimal behavior is not new, the paper considers a substantially more high-level task in comparison to prior work. The paper leverages recent models from the NLP literature which are known to provide strong performance on such question-answering tasks, and is methodologically well grounded in the NLP literature.

(2) The modeling uses text- and question-based features in addition to DNNs, specifically evaluates relevant effects, and compares vanilla pretrained and task-finetuned models. This makes the results more transparent and helps assess the contributions of task optimization. In particular, besides fine-tuned DNNs, the role of the task is further established by directly modeling the question relevance of each word. Specifically, the claim that human reading is predicted better by task-optimized attention distributions rests on (i) a role of question relevance in influencing reading in Expts 1-2 but not 4, and (ii) the fact that fine-tuned DNNs improve prediction of gaze in Expts 1-2 but not 4.

(3) The paper conducts experiments on both L2 and L1 speakers.

Weaknesses:

(1) Under the hypothesis advanced, human reading should adapt rationally to task demands. Indeed, Experiment 1 tests questions from different types in blocks (local and global), and the paper provides evidence that this encourages the development of question-type-specific reading strategies -- indeed, this specifically motivates Experiment 2, and is confirmed indirectly in the comparison of the effects found in the two experiments ("all these results indicated that the readers developed question-type-specific strategies in Experiment 1"). On the other hand, finetuning the model on one of the two types does not seem to reproduce this differential behavior, in the sense that fit to reading data is not improved. In this sense, the model seems to have limited abilities in reproducing the observed task dependence of human reading.

The results support the conclusions well, with the weakness described above a limitation of the modeling approach chosen.

The data are likely to be useful as a benchmark in further modeling of eye-movements, an area of interest to computational research on psycholinguistics.

The modeling results contribute to theoretical understanding of human reading behavior, and strengthens a line of research arguing that it reflects task-adaptive behavior.

---

## [Author Response]

The following is the authors’ response to the original reviews.

**Reviewer #1:**
This manuscript describes a set of four passage-reading experiments which are paired with computational modeling to evaluate how task-optimization might modulate attention during reading. Broadly, participants show faster reading and modulated eye-movement patterns of short passages when given a preview of a question they will be asked. The attention weights of a Transformerbased neural network (BERT and variants) show a statistically reliable fit to these reading patterns above-and-beyond text- and semantic-similarity baseline metrics, as well as a recurrent-networkbased baseline. Reading strategies are modulated when questions are not previewed, and when participants are L1 versus L2 readers, and these patterns are also statistically tracked by the same transformer-based network.I should note that I served as a reviewer on an earlier version of this manuscript at a different venue. I had an overall positive view of the paper at that point, and the same opinion holds here as well.Strengths:Task-optimization is a key notion in current models of reading and the current effort provides a computationally rigorous account of how such task effects might be modeledMultiple experiments provide reasonable effort towards generalization across readers and different reading scenariosUse of RNN-based baseline, text-based features, and semantic features provides a useful baseline for comparing Transformer-based models like BERT

Thank you for the accurate summary and positive evaluation.

Weaknesses:1. Generalization across neural network models seems, to me, somewhat limited: The transformerbased models differ from baseline models in numerous ways (model size, training data, scoring algorithm); it is thus not clear what properties of these models necessarily supports their fit to human reading patterns.

Thank you for the insightful comment. To dissociate the effect of model architecture and the effect of training data, we have now compared the attention weights across three transformer-based models that have the same architecture but different training data/task: randomized (with all model parameters being randomized), pretrained, and fine-tuned models. Remarkably, even without training on any data, the attention weights in randomly initialized models exhibited significant similarity to human attention patterns (Figure. 3A). The predictive power of randomly initialized transformer-based models outperformed that of the SAR model. Through subsequent pre-training and fine-tuning, the predictive capacity of the models was further elevated. Therefore, both model architecture and the training data/task contribute to human-like attention distribution in the transformer models. We have now reported this result:

“The attention weights of randomly initialized transformer-based models could predict the human word reading time and the predictive power, which was around 0.3, was significantly higher than the chance level and the SAR (Fig. 3A, Table S1). The attention weights of pre-trained transformerbased models could also predict the human word reading time, and the predictive power was around 0.5, significantly higher than the predictive power of heuristic models, the SAR, and randomly initialized transformer-based models (Fig. 3A, Table S1). The predictive power was further boosted for local but not global questions when the models were fine-tuned to perform the goal-directed reading task (Fig. 3A, Table S1).”

In addition, we reported how training influenced the sensitivity of attention weights to text features and question relevance. As shown in Figure 4AB, attention in the randomized models were sensitive to text features across all layers. After pretraining, the models exhibited increased sensitivity to text features in the shallow layers, and decreased sensitivity to text features in deep layers. Subsequent finetuning on the reading comprehension task further attenuates the encoding of text features in deep layers but strengthens the sensitivity to task-relevant information.

1. Inferential statistics are based on a series of linear regressions, but these differ markedly in model size (BERT models involve 144 attention-based regressor, while the RNN-based model uses just 1 attention-based regressor). How are improvements in model fit balanced against changes in model size?

Thank you for pointing out this issue. The performance of linear regressions was evaluated based on 5-fold cross-validation, and the performance we reported was the performance on the test set. To match the number of parameters, we have now predicted human attention using the average of all heads. The predictive power of the average head was still significantly higher than the predictive power of the SAR model. We have now reported this result in our revised manuscript:

“For the fine-tuned models, we also predict the human word reading time using an unweighted averaged of the 144 attention heads and the predictive power was 0.3, significantly higher than that achieved by the attention weights of SAR (P = 4 × 10-5, bootstrap).”

Also, it was not clear to me how participant-level variance was accounted for in the modeling effort (mixed-effects regression?) These questions may well be easily remedied by more complete reporting.

In the previous manuscript, the word reading time was averaged across participants, and we did not consider the variance between participants. We have now analyzed eye movements of each participant and used the linear mixed effects model to test how different factors affected human word reading time to account for participantslevel and item-level variances.

“Furthermore, a linear mixed effect model also revealed that more than 85% of the DNN attention heads contribute to the prediction of human reading time when considering text features and question relevance as covariates (Supplementary Results).”

“Supplementary MethodsTo characterize the influences of different factors on human word reading time, we employed linear mixed effects models [5] implemented in the lmerTest package [6] of R. For the baseline model, we treated the type of questions (local vs. global; local = baseline) and all text/task-related features as fixed factors, and considered the interaction between the type of questions and these text/taskrelated features. We included participants and items (i.e., questions) as random factors, each with associated random intercepts…”

Supplementary ResultsThe baseline mixed model revealed significant fixed effects for question type and all text/task-related features, as well as significant interactions between question type and these text/task-related features (Table S7). Upon involving SAR attention, we observed a statistically significant fixed effect associated with SAR attention. When involving attention weights of randomly initialized BERT, the mixed model revealed that most attention heads exhibited significant fixed effects, suggesting their contributions to the prediction of human word reading time. A broader range of attention heads showed significant fixed effects for both pre-trained and fine-tuned BERT.

1. Experiment 1 was paired with a relatively comprehensive discussion of how attention weights mapped to reading times, but the same sort of analysis was not reported for Exps 2-4; this seems like a missed opportunity given the broader interest in testing how reading strategies might change across the different parameters of the four experiments.

Thank you for the valuable suggestion. We have now also characterized how different reading measures, e.g., gaze duration and counts or rereading, were affected by text and task-related features in Experiments 2-4.

For Experiment 2:“For local questions, consistent with Experiment 1, the effects of question relevance significantly increased from early to late processing stages that are separately indexed by gaze duration and counts of rereading (Fig. S9A, Table S3).”

For Experiment 3:“For local questions, the layout effect was more salient for gaze duration than for counts of rereading. In contrast, the effect of word-related features and task relevance was more salient for counts of rereading than gaze duration (Fig. S9B, Table S3).”

For Experiment 4:“Both the early and late processing stages of human reading were significantly affected by layout and word features, and the effects were larger for the late processing stage indexed by counts of rereading (Fig. S9C, Table S3).”

1. Comparison of predictive power of BERT weights to human annotations of text relevance is limited: The annotation task asked participants to chose the 5 "most relevant" words for a given question; if >5 words carried utility in answering a question, this would not be captured by the annotation. It seems to me that the improvement of BERT over human annotations discussed around page 10-11 could well be due to this arbitrary limitation of the annotations.

Thank you for the insightful comment. We only allowed a participant to label 5 words since we wanted the participant to only label the most important information. As the reviewer pointed out, five words may not be enough. However, this problem is alleviated by having >26 annotators per question. Although each participant can label up to 5 words, pooling the results across >26 annotators results in nonzero relevance rating for an average 21.1 words for local questions and 26.1 words for global question. More important, as was outlined in Experimental Materials, we asked additional participants to answer questions based on only 5 annotated keywords. The accuracy for question answering were 75.9% for global questions and 67.6% for local questions, which was close to the accuracy achieved when the complete passage was present (Fig. 1B), suggesting that even 5 keywords could support question answering.

1. Abstract ln 35: This concluding sentence didn't really capture the key contribution of the paper which, at least from my perspective, was something closer to "we offer a computational account of how task optimization modulates attention during reading"p 4 ln 66: I think this sentence does a good job capturing the main contributions of this paper

Thanks for your suggestion. We have modified our conclusion in Abstract accordingly.

1. p 4 ln 81: "therefore is conceptually similar" maybe "may serve a conceptually similar role"

We have rewritten the sentence.

“Attention in DNN also functions as a mechanism to selectively extract useful information, and therefore attention may potentially serve a conceptually similar role in DNN.”

1. p. 7 ln 140: "disproportional to the reading time" I didn't understand this sentence

Sorry for the confusion and we have rewritten the sentence.

“In Experiment 1, participants were allowed to read each passage for 2 minutes. Nevertheless, to encourage the participants to develop an effective reading strategy, the monetary reward the participant received decreased as they spent more time reading the passage (see Materials and Methods for details).”

1. p 8 ln 151: This was another sentence that helped solidify the main research contributions for me; I wonder if this framing could be promoted earlier?

Thank you for the suggestion and we have moved the sentence to Introduction.

1. p. 33: I may be missing something here, but I didn't follow the reasoning behind quantifying model fit against eye-tracking measures using accuracy in a permutation test. Models are assessed in terms of the proportion of random shuffles that show a greater statistical correlation. Does that mean that an accuracy value like 0.3 (p. 10 ln 208) means that 0.7 random permutations of word order led to higher correlations between attention weights and RT? Given that RT is continuous, I wonder if a measure of model fit such as RMSE or even R^2 could be more interpretable.

We have now realized that the term “prediction accuracy” was not clearly defined and have caused confusion. Therefore, in the revised manuscript, we have replaced this term with “predictive power”. Additionally, we have now introduced a clear definition of “prediction power” at its first mention in Result:

“…the predictive power, i.e., the Pearson correlation coefficient between the predicted and real word reading time, was around 0.2”

The permutation test was used to test if the predictive power is above chance. Specifically, if the predictive power is higher than the 95 percentile of the chancelevel predictive power estimated using permutations, the significant level (i.e., the p value) is 0.05. We have explained this in Statistical tests.

1. p. 33: FDR-based multiple comparisons are noted several times, but wasn't clear to me what the comparison set is for any given test; more details would be helpful (e.g. X comparisons were conducted across passages/model-variants/whatever)

Sorry for missing this important information. We have now mentioned which comparisons are corrected,

“…Furthermore, the predictive power was higher for global than local questions (P = 4 × 10-5, bootstrap, FDR corrected for comparisons across 3 features, i.e., layout features, word features, and question relevance)…”

**Reviewer #2:**
In this study, researchers aim to understand the computational principles behind attention allocation in goal-directed reading tasks. They explore how deep neural networks (DNNs) optimized for reading tasks can predict reading time and attention distribution. The findings show that attention weights in transformer-based DNNs predict reading time for each word. Eye tracking reveals that readers focus on basic text features and question-relevant information during initial reading and rereading, respectively. Attention weights in shallow and deep DNN layers are separately influenced by text features and question relevance. Additionally, when readers read without a specific question in mind, DNNs optimized for word prediction tasks can predict their reading time. Based on these findings, the authors suggest that attention in real-world reading can be understood as a result of task optimization.The research question pursued by the study is interesting and important. The manuscript was well written and enjoyable to read. However, I do have some concerns.

We thank the reviewer for the accurate summary and positive evaluation.

1. In the first paragraph of the manuscript, it appears that the purpose of the study was to test the optimization hypothesis in natural tasks. However, the cited papers mainly focus on covert visual attention, while the present study primarily focuses on overt attention (eye movements). It is crucial to clearly distinguish between these two types of attention and state that the study mainly focuses on overt attention at the beginning of the manuscript.

Thank you for pointing out this issue. We have explicitly mentioned that we focus on overt attention in the current study. Furthermore, we have also discussed that native readers may rely more on covert attention so that they do not need to spend more time overtly fixating at the task relevant words.

In Introduction:

“Reading is one of the most common and most sophisticated human behaviors [16, 17], and it is strongly regulated by attention: Since readers can only recognize a couple of words within one fixation, they have to overtly shift their fixation to read a line of text [3]. Thus, eye movements serve as an overt expression of attention allocation during reading [3, 18].”

In Discussion:

“Therefore, it is possible that when readers are more skilled and when the passage is relatively easy to read, their processing is so efficient so that they do not need extra time to encode task-relevant information and may rely on covert attention to prioritize the processing of task-relevant information.”

1. The manuscript correctly describes attention in DNN as a mechanism to selectively extract useful information. However, eye-movement measures such as gaze duration and total reading time are primarily influenced by the time needed to process words. Therefore, there is a doubt whether the argument stating that attention in DNN is conceptually similar to the human attention mechanism at the computational level is correct. It is strongly suggested that the authors thoroughly discuss whether these concepts describe the same or different things.

Thank you for bringing up this very important issue and we have added discussions about why human and DNN may generate similar attention distributions. For example, we found that both DNN and human attention distributions are modulated by task relevance and word properties, which include word length, word frequency, and word surprisal. The influence of task relevance is relatively straightforward since both human readers and DNN should rely more on task relevant words to answer questions. The influence of word properties is less apparent for models than for human readers and we have added discussions:

For DNN’s sensitivity to word surprisal:

“The transformer-based DNN models analyzed here are optimized in two steps, i.e., pre-training and fine-tuning. The results show that pre-training leads to text-based attention that can well explain general-purpose reading in Experiment 4, while the fine-tuning process leads to goal-directed attention in Experiments 1-3 (Fig. 4B & Fig. 5A). Pre-training is also achieved through task optimization, and the pre-training task used in all the three models analyzed here is to predict a word based on the context. The purpose of the word prediction task is to let models learn the general statistical regularity in a language based on large corpora, which is crucial for model performance on downstream tasks [21, 22, 33], and this process can naturally introduce the sensitivity to word surprisal, i.e., how unpredictable a word is given the context.”

For DNN’s sensitivity to word length:

“Additionally, the tokenization process in DNN can also contribute to the similarity between human and DNN attention distributions: DNN first separates words into tokens (e.g., “tokenization” is separated into “token” and “ization”). Tokens are units that are learned based on co-occurrence of letters, and is not strictly linked to any linguistically defined units. Since longer words tend to be separated into more tokens, i.e., fragments of frequently co-occurred letters, longer words receive more attention even if the model pay uniform attention to each of its input, i.e., a token.”

1. When reporting how reading time was predicted by attention weights, the authors used "prediction accuracy." While this measure is useful for comparing different models, it is less informative for readers to understand the quality of the prediction. It would be more helpful if the results of regression models were also reported.

Sorry for the confusion. The prediction accuracy was defined as the correlation coefficient between the predicted and actual eye-tracking measures. We have now realized that the term “prediction accuracy” might have caused confusion. Therefore, in the revised manuscript, we have replaced this term with “predictive power”. Additionally, we have now introduced a clear definition of “prediction power” at its first mention in Result:

“…the predictive power, i.e., the Pearson correlation coefficient between the predicted and real word reading time, was around 0.2”

1. The motivations of Experiments 2 and 3 could be better described. In their current form, it is challenging to understand how these experiments contribute to understanding the major research question of the study.

Thank you for pointing out this issue. In Experiments 1, different types of questions were presented in separate blocks, and all the participants were L2 reader. Therefore, we conducted Experiments 2 and 3 to examine how reading behaviors were modulated when different types of questions were presented in a mixed manner, or when participants were L1 readers. We have now clarified the motivations:

“In Experiment 1, different types of questions were presented in blocks which encouraged the participants to develop question-type-specific reading strategies. Next, we ran Experiment 2, in which questions from different types were mixed and presented in a randomized order, to test whether the participants developed question-type-specific strategies in Experiment 1.”

“Experiments 1 and 2 recruited L2 readers. To investigate how language proficiency influenced task modulation of attention and the optimality of attention distribution, we ran Experiment 3, which was the same as Experiment 2 except that the participants were native English readers.”

**Reviewer #3:**
This paper presents several eyetracking experiments measuring task-directed reading behavior where subjects read texts and answered questions.It then models the measured reading times using attention patterns derived from deep-neural network models from the natural language processing literature.Results are taken to support the theoretical claim that human reading reflects task-optimized attention allocation.STRENGTHS:1. The paper leverages modern machine learning to model a high-level behavioral task (reading comprehension). While the claim that human attention reflects optimal behavior is not new, the paper considers a substantially more high-level task in comparison to prior work. The paper leverages recent models from the NLP literature which are known to provide strong performance on such question-answering tasks, and is methodologically well grounded in the NLP literature.1. The modeling uses text- and question-based features in addition to DNNs, specifically evaluates relevant effects, and compares vanilla pretrained and task-finetuned models. This makes the results more transparent and helps assess the contributions of task optimization. In particular, besides finetuned DNNs, the role of the task is further established by directly modeling the question relevance of each word. Specifically, the claim that human reading is predicted better by task-optimized attention distributions rests on (i) a role of question relevance in influencing reading in Expts 1-2 but not 4, and (ii) the fact that fine-tuned DNNs improve prediction of gaze in Expts 1-2 but not 4.1. The paper conducts experiments on both L2 and L1 speakers.

We thank the reviewer for the accurate summary and positive evaluation.

WEAKNESSES:1. The paper aims to show that human gaze is predicted the the DNN-derived task-optimal attention distribution, but the paper does not actually derive a task-optimal attention distribution. Rather, the DNNs are used to extract 144 different attention distributions, which are then put into a regression with coefficients fitted to predict human attention. As a consequence, the model has 144 free parameters without apparent a-priori constraint or theoretical interpretation. In this sense, there is a slight mismatch between what the modeling aims to establish and what it actually does.Regarding Weakness (1): This weakness should be made explicit, at least by rephrasing line 90. The authors could also evaluate whether there is either a specific attention head, or one specific linear combination (e.g. a simple average of all heads) that predicts the human data well.

Thank you for pointing out this issue. One the one hand, we have now also predicted human attention using the average of all heads, i.e., the simple average suggested by the reviewer. The predictive power of the average head was still significantly higher than the predictive power of the SAR model. We have now reported this result in our revised manuscript.

“For the fine-tuned models, we also predict the human word reading time using an unweighted averaged of the 144 attention heads and the predictive power was 0.3, significantly higher than that achieved by the attention weights of SAR (P = 4 × 10-5, bootstrap).”

On the other hand, since different attention weights may contribute differently to the prediction of human reading time, we have now also reported the weights assigned to individual attention head during the original regression analysis (Fig. S4). It was observed that the weight was highly distributed across attention head and was not dominated by a single head.

Even more importantly, we have now rephrased the statement in line 90 of the previous manuscript:

“We employed DNNs to derive a set of attention weights that are optimized for the goal-directed reading task, and tested whether such optimal weights could explain human attention measured by eye tracking.”

Furthermore, in Discussion, we mentioned that:

“Furthermore, we demonstrate that both humans and transformer-based DNN models achieve taskoptimal attention distribution in multiple steps… Similarly, the DNN models do not yield a single attention distribution, and instead it generates multiple attention distributions, i.e., heads, for each layer. Here, we demonstrate that basic text features mainly modulate the attention weights in shallow layers, while the question relevance of a word modulates the attention weights in deep layers, reflecting hierarchical control of attention to optimize task performance. The attention weights in both the shallow and deep layers of DNN contribute to the explanation of human word reading time (Fig. S4).”

1. While Experiment 1 tests questions from different types in blocks, and the paper mentions that this might encourage the development of question-type-specific reading strategies -- indeed, this specifically motivates Experiment 2, and is confirmed indirectly in the comparison of the effects found in the two experiments ("all these results indicated that the readers developed question-typespecific strategies in Experiment 1") -- the paper seems to miss the opportunity to also test whether DNNs fine-tuned for each of the question-types predict specifically the reading times on the respective question types in Experiment 1. Testing not only whether DNN-derived features can differentially predict normal reading vs targeted reading, but also different targeted reading tasks, would be a strong test of the approach.Regarding Weakness (2): results after finetuning for each question type could be reported.

Thank you for the valuable suggestion. We have now fine-tuned the models separately based on global and local questions. The detailed fine-tuning parameters employed in the fine-tuning process were presented in Author response table 1.

**Author response table 1. sa4table1:** The hyperparameter for fine-tuning DNN models with specific question type.

fold of cross-validation	training :validation : test	learning rate	training epochs	training batchsize	weigh decay	early stoppingpatience
10	70%:20%:10%	1e-5	10	16	0.1	2

The fine-tuning process yielded a slight reduction in loss (i.e., the negative logarithmic score of the correct option) on the validation set. Specifically, for BERT, the loss decreased from 1.08 to 0.96; for ALBERT, it decreased from 1.16 to 0.76; for RoBERTa, it went down from 0.68 to 0.54. Nevertheless, the fine-tuning process did not improve the prediction of reading time (Author response image 1). A likely reason is that the number of global and local questions for training is limited (local questions: 520; global questions: 280), and similar questions also exist in RACE dataset that is used for the original fine tuning (sample size: 87,866). Therefore, a small number of questions can significantly change the reading strategy of human readers but using these questions to effectively fine-tune a model seems to be a more challenging task.

**Author response image 1. sa4fig1:** Fine-tuning based on local and global questions does not significantly modulate the prediction of human reading time. Lighter-color symbols show the results for the 3 BERT-family models (i.e., BERT, ALBERT, and RoBERTa) and the darker-color symbols show the average over the 3 BERT-family models. trans_fine: model fine-tuned based on the RACE dataset; trans_local: models additionally fine-tuned using local questions; trans_global: models additionally fine-tuned using global questions.

1. The paper compares the DNN-derived features to word-related features such as frequency and surprisal and reports that the DNN features are predictive even when the others are regressed out (Figure S3). However, these features are operationalized in a way that puts them at an unfair disadvantage when compared to the DNNs: word frequency is estimated from the BNC corpus; surprisal is derived from the same corpus and derived using a trigram model. The BNC corpus contains 100 Million words, whereas BERT was trained on several Billions of words. Relatedly, trigram models are now far surpassed by DNN-based language models. Specifically, it is known that such models do not fit human eyetracking reading times as well as modern DNN-based models (e.g., Figure 2 Dundee in: Wilcox et al, On the Predictive Power of Neural Language Models for Human Real-Time Comprehension Behavior, CogSci 2020). This means that the predictive power of the word-related features is likely to be underestimated and that some residual predictive power is contained in the DNNs, which may implicitly compute quantities related to frequency and surprisal, but were trained on more data. In order to establish that the DNN models are predictive over and above word-related features, and to reliably quantify the predictive power gained by this, the authors could draw on (1) frequency estimated from the corpora used for BERT (BookCorpus + Wikipedia), (2) either train a strong DNN language model, or simply estimate surprisal from a strong off-the-shelf model such as GPT-2.This concern does not fundamentally cast doubt on the conclusions, since the authors found a clear effect of the task relevance of individual words, which by definition is not contained in those baseline models. However, Figure S3 -- specifically Figure S3C -- is likely to inflate the contribution of the DNN model over and above the text-based features.

Thank you for pointing out these issues. Following the valuable suggestion of the reviewer, we have now (1) computed word frequencies based on BookCorpus and Wikipedia and (2) calculated word surprisal using GPT-2.

“The word features included word length, logarithmic word frequency estimated based on the BookCorpus [62] and English Wikipedia using SRILM [68], and word surprisal estimated from GPT-2 Medium [69].”

These recalculated word frequency and surprisal are correlated with the original measures (word frequency: 0.98; surprisal: 0.59), and the updated results are also closely aligned with those reported in the previous manuscript.

Others:1. How does the statistical modeling take into account that measures are repeated both within the items (same texts read by different subjects) and within the subjects (some subject read multiple texts)? I only see the items-level repetition be addressed in line 715-721 in comparing between local and global questions, but not elsewhere. The standard approach in the literature on human reading times (e.g. the Wilcox et al paper mentioned above, or ref. 44) is to use mixed-effects regression with appropriate random effects for items and subjects. The same question applies to the calculation of chance accuracy (line 702-709), which is done by shuffling words within a passage. Relatedly, how exactly was cross-validation (line 681) calculated? On the level of subjects, individual words, trials, texts, ...?

Thank you for raising up this issue. In the previous manuscript, the word reading time was averaged across participants. The cross-validation was conducted on the level of texts (i.e., passages). Following the valuable suggestion, we have now separately analyzed each participant and applied the linear mixed effects models.

“Furthermore, a linear mixed effect model also revealed that more than 85% of the DNN attention heads contribute to the prediction of human reading time when considering text features and question relevance as covariates (Supplementary Results).”

“Supplementary MethodsTo characterize the influences of different factors on human word reading time, we employed linear mixed effects models [5] implemented in the lmerTest package [6] of R. For the baseline model, we treated the type of questions (local vs. global; local = baseline) and all text/task-related features as fixed factors, and considered the interaction between the type of questions and these text/taskrelated features. We included participants and items (i.e., questions) as random factors, each with associated random intercepts…”

Supplementary ResultsThe baseline mixed model revealed significant fixed effects for question type and all text/task-related features, as well as significant interactions between question type and these text/task-related features (Table S7). Upon involving SAR attention, we observed a statistically significant fixed effect associated with SAR attention. When involving attention weights of randomly initialized BERT, the mixed model revealed that most attention heads exhibited significant fixed effects, suggesting their contributions to the prediction of human word reading time. A broader range of attention heads showed significant fixed effects for both pre-trained and fine-tuned BERT.

1. I could not find any statement about code availability (only about data availability). Will the source code and statistical analysis code also be made available?

We have added the code availability statement.

“The code is now available at https://github.com/jiajiezou/TOA.”

1. The theoretical claim, and some basic features of the research, are quite similar to other recent work (Hahn and Keller, Modeling task effects in human reading with neural network-based attention, Cognition, 2023; cited with very little discussion as ref 44), which also considered task-directed reading in a question-answering task and derived task-optimized attention distributions. There are various differences, and the paper under consideration has both weaknesses and strengths when compared to that existing work -- e.g., that paper derived a single attention distribution from task optimization, but the paper under consideration provides more detailed qualitative analysis of the task effects, uses questions requiring more high-level reasoning, and uses more state-of-the-art DNNs.Thanks for bringing up this issue. We have now incorporated a more comprehensive discussion that compare the current study with the recent work conducted by Hahn and Keller:

“When readers read a passage to answer a question that can be answered using a word-matching strategy [45], a recent study has demonstrated that the specific reading goal modulates the word reading time and the effect can be modeled using a RNN model [46]. Here, we focus on questions that cannot be answered using a word-matching strategy (Fig. 1B) and demonstrate that, for these challenging questions, attention is still modulated by the reading goal but the attention modulation cannot be explained by a word-matching model (Fig. S3). Instead, the attention effect is better captured by transformer models than an advanced RNN model, i.e., the SAR (Fig. 3A). Combining the current study and the study by Hahn et al. [46], it is possible that the word reading time during a general-purpose reading task can be explained by a word prediction task, the word reading time during a simple goal-directed reading task that can be solved by word matching can be modeled by a RNN model, while the word reading time during a more complex goal-directed reading task involving inference is better modeled using a transformer model. The current study also further demonstrates that elongated reading time on task-relevant words is caused by counts of rereading and further studies are required to establish whether earlier eye movement measures can be modulated by, e.g., a word matching task.”

1. In Materials&Methods, line 599-636, specifically when "pretraining" is mentioned (line 632), it should be mentioned what datasets these DNNs were pretrained on.

We have now mentioned this in the revised manuscript:

“The pre-training process aimed to learn general statistical regularities in a language based on large corpora, i.e., BooksCorpus [62] and English Wikipedia…”